# THINK TWICE: BRANCH-AND-RETHINK REASONING REWARD MODEL

## ABSTRACT

Large language models (LLMs) increasingly rely on thinking models that externalize intermediate steps and allocate extra test-time compute, with *think-twice* strategies showing that a deliberate second pass can elicit stronger reasoning. In contrast, most reward models (RMs) still compress many quality dimensions into a single scalar in one shot, a design that induces *judgment diffusion*: attention spreads across evaluation criteria, yielding diluted focus and shallow analysis. We introduce **branch-and-rethink** (BR-RM), a two-turn RM that transfers the think-twice principle to reward modeling. Turn 1 performs *adaptive branching*, selecting a small set of instance-critical dimensions (such as factuality and safety) and sketching concise, evidence-seeking hypotheses. Turn 2 executes *branch-conditioned rethinking*, a targeted reread that tests those hypotheses and scrutinizes only what matters most. We train with GRPO-style reinforcement learning over structured two-turn traces using a simple binary outcome reward with strict format checks, making the approach compatible with standard RLHF pipelines. By converting all-at-once scoring into focused, second-look reasoning, BR-RM reduces judgment diffusion and improves sensitivity to subtle yet consequential errors while remaining practical and scalable. Experimental results demonstrate that our model achieves state-of-the-art performance on three challenging reward modeling benchmarks across diverse domains.

## 1 INTRODUCTION

Large language models (LLMs) increasingly solve hard problems by *reasoning* rather than merely responding (Shao et al., 2024; DeepSeek-AI et al., 2025). Under the banner of "reasoning LMs," models externalize intermediate steps – e.g., chain-of-thought, scratchpads, and reflect-revise traces – and allocate more test-time compute to difficult instances (Wei et al., 2022; Nye et al., 2021; Shinn et al., 2023; Yao et al., 2023; Wang et al., 2023). These procedures convert raw capacity into usable reasoning: they surface hypotheses, check local inferences, and expose mistakes that would otherwise remain hidden. Recent "thinking" models and analyses of test-time compute further highlight the benefits of deliberate, multi-step inference and adaptive allocation (Snell et al., 2024).

Yet a single explicit-reasoning pass is often brittle – a quiet factual slip, a local contradiction, or a small logic bug can persist because attention is spread across many potential concerns. Intuitively and empirically, thinking twice helps: a second, targeted pass re-reads with hypotheses, directs scrutiny to suspicious spans, and reallocates compute where uncertainty is highest (Wang et al., 2023; Tian et al., 2025). This naturally raises the question: if solvers benefit from a second look, should our *judges* – that grade model outputs – be designed to do the same?

Enter Reward Models (RMs), the workhorse that turns preference signals into scalar rewards in RLHF- and RLAIF-style pipelines (Ouyang et al., 2022; Bai et al., 2022b). *Scalar RMs* typically output a single holistic score per response – implicitly aggregating all aspects of response quality in a single forward pass. Because it asks the judge to consider everything at once, it discourages deep scrutiny of what matters most for a given instance.

To address this limitation, *Generative Reward Models* (GenRMs) ask the judge to explain before scoring: the RM first produces a short critique or rationale and then outputs a number, improving transparency and often robustness (Mahan et al., 2024; Zhang et al., 2025; Ye et al., 2025). Still, most GenRMs reason under a fixed rubric and ultimately collapse their critique into a single global

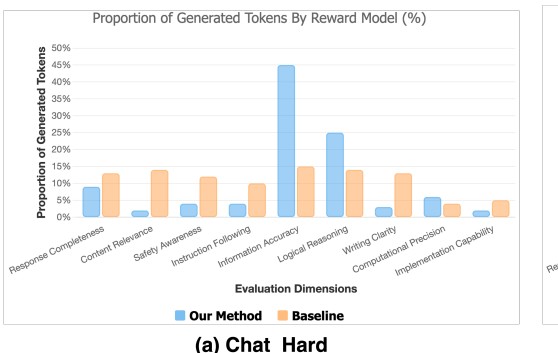 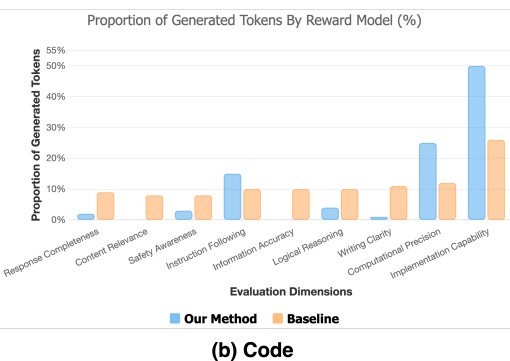

(a) Chat_Hard
(b) Code

Figure 1: Comparison of token allocation between our method and a recent ReasonRM (Guo et al., 2025) on two subsets from RM-Bench. Our method adaptively focuses its generative analysis on a few critical dimensions for each task (e.g., Information Accuracy for chat), while the baseline shows focus dilution by spreading its tokens broadly across many criteria. More details in Appendix A.

decision, so the underlying all-at-once pressure remains. Based on this, *Reasoning Reward Models* (ReasonRMs) treat judgment itself as a reasoning task, deliberately allocating test-time compute prior to scoring (Chen et al., 2025b; Guo et al., 2025). Yet ReasonRMs typically reason broadly (rubric-wide) and do not enforce instance-adaptive focus or a second pass conditioned on discovered issues. Thus, even as they add reasoning, they do not guarantee that attention concentrates where risk is highest.

We highlight a recurring weakness across existing RMs, which we call *judgment diffusion*. When forced to grade responses holistically, these models spread their attention thin across many possible criteria. The result is **focus dilution**, where no single dimension is examined with sufficient depth, and **shallow analysis**, where judgments skim the surface instead of probing concrete failure modes. Empirical studies reinforce this view: RMs and LLM judges often miss subtle factual slips or localized reasoning errors, and their decisions are easily swayed by stylistic factors rather than substantive quality (Lambert et al., 2025a; Liu et al., 2025b). To empirically ground our hypothesis, we conduct a preliminary study of the generation from a recent ReasonRM (Guo et al., 2025) on two challenging subsets of RM-Bench (Liu et al., 2025b). We analyze how this model allocates its generative token budget across different evaluation dimensions. Figure 1 shows this model spreads its analysis thinly across multiple dimensions without detailed reasoning on the most critical dimensions.

Motivated by these observations, we introduce a two-turn reward modeling framework that builds *focus* and *rethinking* into the RM. Instead of grading everything at once, our framework recasts judgment as **branch-and-rethink**: the model first narrows to a few instance-critical dimensions, then performs a second, conditioned pass that scrutinizes only what matters most. Specifically, the first turn is **Adaptive Branching**. The RM selects a few relevant cognitive dimensions from nine candidates (e.g., factual accuracy, reasoning quality, safety) and produces a concise issue sketch pointing to potential weaknesses. This focuses attention where risk is highest, replacing diffusion with targeted scrutiny. The second turn is **Branch-Conditioned Rethinking**. Using the first-turn findings, RM re-reads the response through the lens of flagged dimensions – e.g., verifying facts, checking brittle reasoning, or examining localized bugs. This issue-driven pass avoids the common failure mode of broad but shallow reasoning. For training, we adopt GRPO-style reinforcement learning to support two-turn traces: both turns produce structured outputs, and optimization uses a simple binary outcome reward with strict format checks. This keeps supervision clean, interoperates with standard RLHF infrastructure, and ensures compute is spent where a second look matters.

Extensive experiments on reward modeling benchmarks show that our method, BR-RM, achieves state-of-the-art performance consistently on RewardBench (Lambert et al., 2025b), RMBench (Liu et al., 2025b), and RMB (Zhou et al., 2025), outperforming strong baselines across multiple domains, including reasoning, general knowledge, safety, and alignment with human preference. Beyond final performance, we conduct extensive empirical analyses of BR-RM, including the ablation study of our training recipes, the detailed analysis of the reward designs, and concrete impacts of training data sources. These insights provide a deeper understanding of reward reasoning processes and will hopefully inspire the development of future reward reasoning models.

## 2 RELATED WORK

### 2.1 REASONING LANGUAGE MODELS

Reasoning has emerged as a defining capability of modern LLMs, marking a shift from surface-level response generation to structured problem-solving. Early advances such as chain-of-thought prompting, scratchpads, tree-structured exploration, and self-consistency demonstrate that explicitly externalizing intermediate steps improves accuracy and interpretability in multi-step tasks (Wei et al., 2022; Nye et al., 2021; Yao et al., 2023; Wang et al., 2023). By surfacing hypotheses and local inferences, these methods transform raw capacity into usable reasoning, enabling models to identify and correct mistakes that would otherwise remain hidden. Beyond single-pass prompting, reflect–revise and multi-pass paradigms show that "thinking twice" helps models correct quiet factual slips and local logic bugs through targeted second looks (Shinn et al., 2023; Snell et al., 2024; Tian et al., 2025). Domain-focused reasoning systems (e.g., math) further highlight that deliberate computation and reinforcement learning can incentivize deeper reasoning traces (Shao et al., 2024; DeepSeek-AI et al., 2025). Collectively, these works suggest that allocating test-time compute and conditioning a second pass on suspected weaknesses can meaningfully improve reliability. We draw on this intuition, transferring the "think twice" principle from solvers to judges by enforcing a focused, issue-conditioned second pass in our framework.

### 2.2 REWARD MODELS

Reward models (RMs) are central to alignment pipelines such as RLHF, converting human or AI preference signals into scalar rewards for optimization (Ouyang et al., 2022; Bai et al., 2022b). Classical discriminative RMs predict a single holistic score per response, implicitly aggregating in a single forward pass factuality, reasoning quality, safety, coding correctness, instruction following, style, etc. (Christiano et al., 2017; Ziegler et al., 2019; Stiennon et al., 2020; Bai et al., 2022a; Sun et al., 2023; Liu et al., 2024a; Zhu et al., 2024). While simple and scalable, this all-at-once design dilutes attention across dimensions, limiting sensitivity to subtle but consequential errors.

To improve transparency and robustness, Generative Reward Models (GenRMs) introduce *explain-then-score*: the model produces a short critique or rationale prior to emitting a score (Mahan et al., 2024; Zhang et al., 2025; Ankner et al., 2024; Liu et al., 2025a). However, most systems still collapse critiques to a single global decision and rarely enforce instance-adaptive focus. Recent Reasoning Reward Models (ReasonRMs) treat judgment itself as a reasoning task, allocating test-time compute prior to scoring (Chen et al., 2025b; Guo et al., 2025; Whitehouse et al., 2025). Yet many reason broadly over fixed rubrics, without guaranteeing that scrutiny concentrates on the instance-critical dimensions discovered during evaluation. DeepSeek-GRM (Liu et al., 2025c) is a notable exception, prompting the model to generate instance-specific evaluation criteria, but does so in a single-turn, free-form manner and still maintains a fixed general rubric. In contrast, we found our structured two-turn "branch-and-rethink" approach to be crucial to reduce judgment diffusion and increase accuracy. Similar in spirit to our work, EvalPlanner (Saha et al., 2025) trains a judge model that first plans detailed evaluation steps before executing on them: the generated plan, however, is entirely open-ended – while we constrain the model to choose among specific criteria to focus on – and the approach relies on iterative Direct Preference Optimization (Rafailov et al., 2023) – while we use online Reinforcement Learning.

## 3 METHOD

Our approach, **branch-and-rethink**, reframes reward modeling from a direct regression task over a scalar value to a structured reasoning and generation task. We introduce a framework that compels the reward model to produce an explicit, structured deliberation trace. This is achieved through a sequential, two-stage generation process designed to mitigate the *judgment diffusion* observed in single-pass evaluators, by first establishing a focused analytical scope and then performing a deep, conditioned analysis. Fig. 2 contrasts our method with generative RMs and general Reasoning RMs.

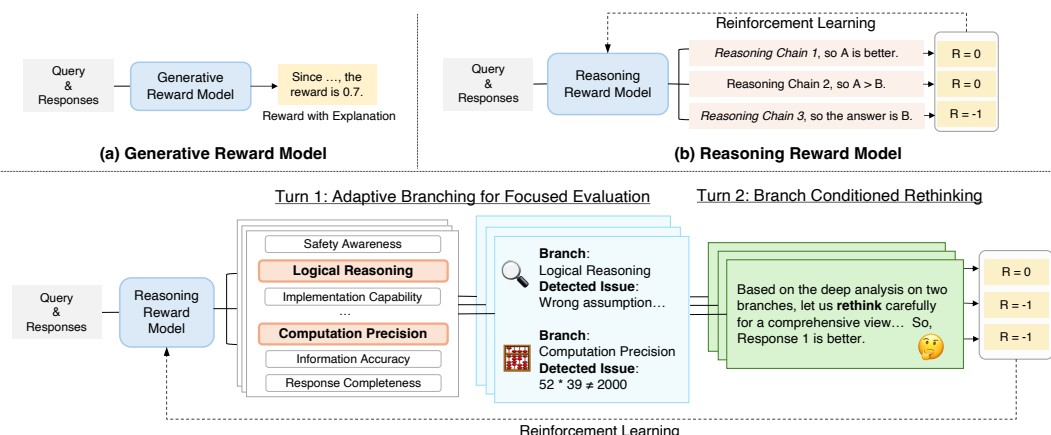

Figure 2: Illustration of our proposed method, comparing with GenRM and ReasonRM. Our Branch-and-Rethink Reasoning Reward Model first performs Turn 1: Adaptive Branching, where it selects a few critical dimensions (e.g., Logical Reasoning, Computation Precision) to focus its evaluation and detect specific issues. This focused analysis then informs Turn 2: Branch-Conditioned Rethinking, where the model conducts a deeper, issue-driven re-thinking to arrive at a final reward judgment, which is then used for reinforcement learning.

### 3.1 TASK FORMULATION

The standard reward modeling task is to learn a preference function from a dataset $\mathcal{D} = \{(x_i, y_{i,1}, y_{i,2}, z_i)\}_{i=1}^N$. For a given prompt $x_i$, the goal is to correctly identify the preferred response between two candidates, $y_{i,1}$ and $y_{i,2}$, as indicated by a human label $z_i \in \{1, 2\}$.

Conventional approaches learn a scalar function $R_\phi(x, y)$ that assigns an implicit score to a response $y$ to prompt $x$. We instead train a policy $\pi_\theta$ to generate a **two-turn deliberation trace**, denoted by $\tau$, for each comparison triplet $(x, y_1, y_2)$. This trace externalizes the model's judgment, culminating in a final preference $\hat{z} \in \{1, 2\}$ that can be extracted from the generated text. The objective is to train $\pi_\theta$ to produce traces where the extracted decision $\hat{z}$ aligns with the ground-truth label $z$.

### 3.2 THE BRANCH-AND-RETHINK FRAMEWORK

To produce the reasoning trace $\tau$, we employ a two-stage generative workflow. This architecture is the core of our method, designed to systematically address the failure modes of *focus dilution* and *shallow analysis*. The complete trace is a concatenation of the outputs from each stage: $\tau = \tau_1 \circ \tau_2$.

#### 3.2.1 STAGE 1: ADAPTIVE BRANCHING FOR FOCUSED EVALUATION

The first stage directly targets **focus dilution**. To prevent the model's evaluation from becoming diffuse, we constrain it to first select a small subset of critical evaluation criteria, $\mathcal{C}_{\text{sel}}$, from a predefined universal set of criteria, $\mathcal{C}$. For example, $\mathcal{C}$ could be {*Factual Accuracy, Logical Coherence, Instruction Adherence, Creativity...*}. This selection forces the model to hypothesize the most salient dimensions for the given comparison. Conditioned on this selection, the model generates a preliminary analysis, $\alpha_j$, for each response $y_j$:

$$(\mathcal{C}_{\text{sel}}, \alpha_1, \alpha_2) \sim \pi_\theta(\cdot \mid x, y_1, y_2) \tag{1}$$

The output of this stage, which includes the selected criteria and the initial analysis, constitutes the first part of the trace, $\tau_1 = (\mathcal{C}_{\text{sel}}, \alpha_1, \alpha_2)$.

#### 3.2.2 STAGE 2: CONDITIONED RETHINKING FOR DEEP ANALYSIS

The second stage is designed to prevent **shallow analysis**. With the analytical scope established by $\tau_1$, the model performs a targeted re-evaluation. This **Branch-Conditioned Rethinking** is not an independent assessment but a deliberate critique of the responses *through the lens* of the criteria

identified in Stage 1. To ensure this deliberation is principled, we incorporate task-specific evaluation hierarchies, $\mathcal{H}_{\text{task}}$, which provide a structured inductive bias. These hierarchies encode a stable ordering of importance for different criteria, such as:

- **Accuracy-Critical Tasks**: Correctness > Methodology > Clarity
- **Creative Tasks**: Intent Alignment > Quality > Novelty

The model is conditioned on $\tau_1$ and guided by $\mathcal{H}_{\text{task}}$ to generate the second part of the trace, $\tau_2$, which contains a detailed comparative judgment and the final decision, $\hat{z}$.

$$\tau_2 \sim \pi_\theta(\cdot \mid x, y_1, y_2, \tau_1) \tag{2}$$

The final verdict, $\hat{z}$, is then extracted from $\tau_2$.

### 3.3 LEARNING VIA REINFORCEMENT LEARNING

#### 3.3.1 THE GRPO TRAINING ALGORITHM

We employ Generalized Reward Policy Optimization (GRPO, Shao et al. 2024), a variant of PPO, chosen for its training stability and sample efficiency. The training loop (Algorithm 1) optimizes the policy $\pi_\theta$ to maximize a composite reward function over the distribution of generated traces.

---

**Algorithm 1** Two-Turn RL Training with GRPO

---

1: **for** each training step **do**
2:     $B \leftarrow \text{SampleBatch}(\mathcal{D}, \text{batch\_size})$     ▷ *Sample a batch of data*
3:     $U \leftarrow \emptyset$     ▷ *Initialize an empty update buffer*
4:     **for** each $(x_i, y_{i,1}, y_{i,2}, z_i) \in B$ **do**
5:         $\tau_{i,1} \leftarrow \pi_\theta(\cdot|x_i, r_{i,1}, r_{i,2}, p_1)$     ▷ *Generate the branching trace*
6:         $\tau_{i,2} \leftarrow \pi_\theta(\cdot \mid x_i, y_{i,1}, y_{i,2}, \tau_{i,1})$     ▷ *Generate the rethinking trace*
7:         $r_i \leftarrow \text{CalculateReward}(z_i, \tau_{i,2})$     ▷ *Calculate reward based on the final decision*
8:         $U \leftarrow U \cup \{(x_i, r_{i,1}, r_{i,2}, \tau_{i,1}, r_i)\}$     ▷ *Add first turn to buffer with reward $r_i$*
9:         $U \leftarrow U \cup \{(x_i, r_{i,1}, r_{i,2}, \tau_{i,1}, \tau_{i,2}, r_i)\}$     ▷ *Add second turn to buffer with same reward $r_i$*
10:     **end for**
11:     $\text{UpdatePolicy}(\pi_\theta, U)$     ▷ *Update policy using the GRPO objective*
12: **end for**

---

The GRPO objective balances reward maximization with training stability via a clipped surrogate objective and a KL-divergence penalty against a reference policy $\pi_{\text{ref}}$:

$$L_{\text{GRPO}} = \mathbb{E}_{\tau \sim \pi_\theta}\left[\min\left(\rho(\tau)A(\tau), \text{clip}(\rho(\tau), 1-\epsilon, 1+\epsilon)A(\tau)\right)\right] - \beta D_{\text{KL}}(\pi_\theta \| \pi_{\text{ref}}) \tag{3}$$

where $\rho(\tau) = \frac{\pi_\theta(\tau)}{\pi_{\text{ref}}(\tau)}$ is the probability ratio. Here $A(\tau)$ is a group-relative, critic-free advantage: for each prompt we sample $K$ traces, compute a terminal reward $r(\tau)$ per trace, subtract the per-prompt mean (and optionally divide by the per-prompt standard deviation), and use this centered (whitened) value as $A(\tau)$. We assign this scalar uniformly to all tokens in the trace.

#### 3.3.2 REWARD FUNCTION DESIGN

Our reward $R(\tau)$ aims to guide the learning process through a curriculum, by summing two terms:

$$R(\tau) = R_{\text{format}}(\tau) + R_{\text{outcome}}(\tau) \tag{4}$$

**Format Reward**: To enforce the two-stage format, we heavily penalize any format deviation.

$$R_{\text{format}}(\tau) = \begin{cases} 0 & \text{if } \tau \text{ is well-formed} \\ \lambda_{\text{format}} & \text{if } \tau \text{ is malformed} \end{cases} \tag{5}$$

The penalty constant $\lambda_{\text{format}} < 0$ ensures that learning the correct generative format is prioritized.

**Binary Outcome Reward**: The substantive quality of a trace is rewarded based on the correctness of its final decision. This reward is only applied if the trace is structurally valid, i.e. $R_{\text{format}}(\tau) = 0$.

$$R_{\text{outcome}}(\tau) = \begin{cases} 0 & \text{if } \hat{z} = z \\ -1 & \text{if } \hat{z} \neq z \end{cases} \tag{6}$$

This binary reward (calculated only after the full trace is generated) encourages the model to produce a reasoning that leads to the correct outcome. The same reward $R(\tau)$ is assigned separately to the first turn (generation of the branching trace $\tau_1$) and the second turn (generation of the rethinking trace $\tau_2$ conditioned on $\tau_1$). This formulation makes it easy to integrate into standard implementations of the GRPO objective without requiring support for multi-turn Reinforcement Learning.

## 4 EXPERIMENT

### 4.1 EXPERIMENTAL SETUP

**Training and Evaluation Data.** To train the proposed model, we used a diverse mix of preference datasets, including HelpSteer3 (Wang et al., 2025), Skywork Reward Preference-80K (Liu et al., 2024b), Code-Preference-Pairs (Vezora, 2024), and Math-Step-DPO-10K (Lai et al., 2024). Model performance was measured by preference accuracy on comprehensive evaluation benchmarks like RewardBench (Lambert et al., 2025b), RM-Bench (Liu et al., 2025b), and RMB (Zhou et al., 2025).

**Baselines.** The baseline models were chosen to cover the main three categories of RMs. The first, Scalar RMs, directly output a score (Yuan et al., 2025; Cai et al., 2024; Liu et al., 2024b; Adler et al., 2024; Chen et al., 2025b). The second category, Generative RMs, produce critiques, features models (Anthropic, 2024; Dubey et al., 2024; Reid et al., 2024; OpenAI, 2023; Liu et al., 2024b). Here GenRMs contains two sub-types, fine-tuned GenRMs and strong general-prupose LLMs via prompting. Lastly, the third type consists of advanced Reasoning RMs based on thinking models (Liu et al., 2025c; Wang et al., 2024; Whitehouse et al., 2025; Saha et al., 2025; Chen et al., 2025b).

**Implementation Details.** Our implementation leverages a scalable and efficient post-training library, NeMo-RL[1], with a custom two-turn reward modeling environment. Regarding model design, the penalty constant for format checking is -100. For training, model-specific learning rates were tuned based on scale: $5 \times 10^{-7}$ for 8B models and $1 \times 10^{-6}$ for 14B models, all trained for 400 optimization steps with validation every 5 steps. For inference, we employ a consistent evaluation protocol with training parameters, computing accuracy as the primary metric, with additional domain-specific metrics for comprehensive analysis. More configurations are in Appendix D.

### 4.2 MAIN RESULTS

Table 1 presents a comprehensive performance comparison of our Branch-and-Rethink Reward Models (BR-RMs) against the best-performing baselines across three standard benchmarks.

**ScalarRMs remain competitive but plateau.** ScalarRMs such as INF-ORM-Llama3.1-70B achieve the strongest RewardBench score (95.1) and lead most scalar baselines in average accuracy (78.8). However, despite high scores on RewardBench, they lag behind reasoning-focused models on RM-Bench (70.9) and RMB (70.5). Smaller ScalarRMs perform relatively well, suggesting that simply scaling up parameters without reasoning mechanisms brings diminishing returns.

**GenRMs improve robustness but trade off consistency.** Generative reward models, which critique before scoring, close some gaps but do not consistently surpass scalar baselines. For example, Skywork-Critic-Llama-3.1-70B excels on RewardBench (93.3), yet underperforms on RMB (65.6). GPT-4o achieves balanced results, with the second-best RMB score (73.8) and an average of 77.7, but still lags behind leading scalars and reasoning RMs.

**ReasonRMs show clear gains, especially at scale.** The RM-R1 model family and related ReasonRMs outperform both ScalarRMs and GenRMs once scaled beyond 14B parameters. RM-R1-DeepSeek-Distilled-Qwen-32B achieves 83.9 on RM-Bench, the strongest baseline prior to our

---

[1]https://github.com/NVIDIA-NeMo/RL

Table 1: The performance comparison between best-performing baselines. Bold numbers indicate the best performance, Underlined numbers indicate the second best. The more detailed numbers on RewardBench, RM-Bench, and RMB are in Table 6, Table 7, and Table 8 in the appendix.

| Model | RewardBench | RM-Bench | RMB | Average |
|---|---|---|---|---|
| **ScalarRMs** | | | | |
| SteerLM-RM-70B | 88.8 | 52.5 | 58.2 | 66.5 |
| Eurus-RM-7b | 82.8 | 65.9 | 68.3 | 72.3 |
| Internlm2-20b-reward | 90.2 | 68.3 | 62.9 | 73.6 |
| Skywork-Reward-Gemma-2-27B | 93.8 | 67.3 | 60.2 | 73.8 |
| Internlm2-7b-reward | 87.6 | 67.1 | 67.1 | 73.9 |
| ArmoRM-Llama3-8B-v0.1 | 90.4 | 67.7 | 64.6 | 74.2 |
| Nemotron-4-340B-Reward | 92.0 | 69.5 | 69.9 | 77.1 |
| Skywork-Reward-Llama-3.1-8B | 92.5 | 70.1 | 69.3 | 77.5 |
| INF-ORM-Llama3.1-70B | **95.1** | 70.9 | 70.5 | 78.8 |
| **GenRMs** | | | | |
| Claude-3.5-sonnet-20240620 | 84.2 | 61.0 | 70.6 | 71.9 |
| Llama3.1-70B-Instruct | 84.0 | 65.5 | 68.9 | 72.8 |
| Gemini-1.5-pro | 88.2 | 75.2 | 56.5 | 73.3 |
| Skywork-Critic-Llama-3.1-70B | 93.3 | 71.9 | 65.5 | 76.9 |
| GPT-4o-0806 | 86.7 | 72.5 | 73.8 | 77.7 |
| **ReasonRMs** | | | | |
| JudgeLRM-7B | 75.2 | 64.7 | 53.1 | 64.3 |
| DeepSeek-PairRM-27B | 87.1 | – | 58.2 | – |
| DeepSeek-GRM-27B | 86.0 | – | 69.0 | – |
| Self-taught-evaluator-Llama3.1-70B | 90.2 | 71.4 | 67.0 | 76.2 |
| J1-Llama-70B | 93.3 | 82.7 | 67.3 | 81.1 |
| EvalPlanner-Llama-3.3-70B-Instruct | 93.8 | 82.1 | – | – |
| RM-R1-Qwen-Instruct-14B | 88.2 | 76.1 | 69.2 | 77.8 |
| RM-R1-DeepSeek-Distilled-Qwen-14B | 88.9 | 81.5 | 68.5 | 79.6 |
| RM-R1-Qwen-Instruct-32B | 91.4 | 79.1 | 73.0 | 81.2 |
| RM-R1-DeepSeek-Distilled-Qwen-32B | 90.9 | 83.9 | 69.8 | 81.5 |
| **Our Methods** | | | | |
| BR-RM-Qwen-8B | 91.0 | 85.0 | 71.8 | **82.6** |
| BR-RM-Qwen-14B | 92.1 | **85.9** | **74.7** | **84.2** |

method, with a solid average of 81.5. Similarly, RM-R1-Qwen-Instruct-32B attains 81.2 average. These results confirm that structured reasoning pipelines – when scaled – are effective at overcoming shallow analysis and improving sensitivity to subtle reasoning flaws. However, smaller reasoning RMs (e.g., JudgeLRM-7B at 64.3 average) trail behind even the weaker ScalarRMs, suggesting that naive reasoning without sufficient scale is not sufficient.

**BR-RM models consistently perform well across all three benchmarks.** While ScalarRMs dominate RewardBench and ReasonRMs lead RM-Bench, our BR-RMs achieve the best overall balance. BR-RM-Qwen-14B reaches 92.1 on RewardBench (close to scalar SOTA), 85.9 on RM-Bench (highest overall), and 74.7 on RMB (top-2 overall), yielding the best average (84.2). Even BR-RM-Qwen-8B, with only 8B parameters, surpasses larger baselines such as GPT-4o and INF-ORM-Llama3.1-70B in average score. These results highlight adaptive two-turn reasoning not only closes the gap with scalar models on factual benchmarks but also sets a new bar for reasoning robustness.

## 4.3 ABLATION STUDY

To validate the contributions of our proposed components and training data mixture, we conduct several ablations based on our BR-RM models. We systematically remove key architectural components to isolate their impact on performance across our evaluation benchmarks (Table 2).

**Branching Only** In this variant, we remove the second "rethinking" turn. This ablation caused the most significant performance degradation. The first turn is designed for both branching and error detection, which directs its focus toward identifying specific, granular issues. Without the second turn, the model's final judgment relies entirely on this initial pass. This can mislead the model to over-emphasize details and minor flaws without having a comprehensive view of the whole response. The absence of the holistic re-evaluation step is particularly damaging on nuanced benchmarks like

Table 2: **Ablation Study.** This table summarizes the performance impact of removing key components and training datasets from our BR-RM models. Values in parentheses show the performance drop from the full models, highlighting the contribution of each component.

| Model | BR-RM-Qwen-8B | | | BR-RM-Qwen-14B | | |
|---|---|---|---|---|---|---|
| | RewardBench | RMBench | RMB | RewardBench | RMBench | RMB |
| **BR-RM (Full Model)** | **91.0** | **85.0** | **71.8** | **92.1** | **85.9** | **74.7** |
| Branching Only | 90.1 | 82.6 | 67.4 | 90.9 | 83.2 | 71.0 |
| Unconditioned Rethink | 90.6 | 83.9 | 69.3 | 91.3 | 84.4 | 71.9 |
| Single-Turn Model | 90.4 | 84.5 | 70.0 | 91.3 | 85.2 | 73.3 |

RMB, where the 14B model's performance drops by a substantial 3.7 points. This shows that the second turn is critical for contextualizing initial findings.

**Unconditioned Rethinking** Here, we retain the two-turn training but remove branching. The first turn goes through all the dimensions while the second turn is prompted with a generic "re-evaluate the response" instruction. While this model performs better than the "Branching Only" version, it remained well below our full model. For instance, the 14B model sees performance drops of 0.8, 1.5, and 2.8 points on the respective benchmarks. This result underscores the importance of adaptive focus. Without an explicit directive from the first turn, the second pass is less efficient and prone to the same judgment diffusion it is designed to prevent.

**Single-Turn Model** To test the value of our sequential process, we merge the prompts for both turns into a single instruction. This approach forces the model to rely on only one-time thinking. The resulting prompt is highly complex, demanding that the model handle too many subtasks at the same time: select dimensions, sketch issues, and then re-evaluate to produce a final score. This cognitive overload prevents the model from deeply engaging with any single subtask, leading to a shallower analysis. The consistent, albeit smaller, performance drop confirms that a deliberate, sequential process is critical for high-quality, reasoned evaluation.

## 4.4 REWARD DESIGN ANALYSIS

Table 3: **Reward Analysis.** Quantitative comparison of our proposed reward design against several ablations. Our approach using a direct binary reward consistently achieves the highest scores. Due to space constraints, RewardBench is abbreviated as RewardB.

| Reward | BR-RM-Qwen-8B | | | BR-RM-Qwen-14B | | |
|---|---|---|---|---|---|---|
| | RewardB. | RMBench | RMB | RewardB. | RMBench | RMB |
| **Our Reward** | **91.0** | **85.0** | **71.8** | **92.1** | **85.9** | **74.7** |
| Removing Format Checks | 90.5 | 84.2 | 71.5 | 91.3 | 85.4 | 73.1 |
| Scoring on a Scale | 90.0 | 83.5 | 70.1 | 90.8 | 85.1 | 71.7 |
| Additional Reward for Branch | 86.5 | 79.0 | 66.2 | 87.8 | 82.7 | 69.3 |

To identify the best training strategy, we evaluated several reward designs in Table 3.

**Removing Format Checks** First, we tested removing the strict formatting that guides the model's two-step reasoning process. While this simplified the setup, it allowed the model to find shortcuts, such as producing minimal text just to get the final answer right while skipping the intended reasoning steps. We found that the format check provides crucial guidance and control. It helps the model distinguish between a wrong answer and a badly formatted one; without it, the model would receive the same penalty for incorrect reasoning or just for messy output, which slows down learning. **Format checks provide essential structure and unambiguous feedback.**

**Scoring on a Scale** In another experiment, we had the model output rate a detailed score on a scale, like from -3 to 3. This is inspired by the reward dataset, HelpSteer3, with fine-grained human-annotated groundtruth to show the preference degree. During training, its reward was based on the distance between its predicted score and a ground-truth score. For training data processing, beyond Helpsteer3 naturally with these fine-grained scores, for datasets with only binary labels, we had to artificially map the winning response to +2 and the losing one to -2. This approach failed due to a

disconnect between the training task and the final evaluation. While training rewarded the model for predicting an exact score, our ultimate evaluation only cared about whether the model correctly identified the winning response (a simple binary choice). This meant the model had no strong incentive to learn the true meaning behind the scores; it only needed to learn if a score was positive or negative. **Predicting precise scores is beneficial when only the winner matters.**

**Additional Reward for Branching** Finally, we tried to reward the first step of the model's reasoning process individually using a different reward. The key issue here is no natural signal to serve for this reward signal unlike the final correct answer (the ground truth). Instead, it was calculated from intermediate values generated during the process. Specifically, we measured the difference of the rewards with and without branching to quantify the effect of Turn 1. But we found this indirect feedback could confuse the model, and it led to two major practical problems: 1) the approach was computationally expensive, nearly doubling the processing required, and 2) the resulting reward signal was infrequent and erratic, which destabilized training. **Calculated intermediate rewards created a noisy, unstable signal.**

**The Best Reward: Binary Reward with Format Checking.** After testing these alternatives, we confirmed that the best reward design is the most straightforward one. In this setup, the model produces a binary choice (A or B) within a required format and receives a simple reward based on whether it is correct. Each of the other designs introduced a flaw – confusing signals, misaligned objectives, or indirect feedback – that hurt performance. This shows that a direct alignment between the task and a reward based on the correct answer provides the most effective foundation for training.

### 4.5 TRAINING DATA ANALYSIS

Table 4: **Training Data Analysis.** Impact of training data sources for our BR-RM models.

| Model | BR-RM-Qwen-8B | | | BR-RM-Qwen-14B | | |
|---|---|---|---|---|---|---|
| | RewardBench | RMBench | RMB | RewardBench | RMBench | RMB |
| **Full training data** | **91.0** | **85.0** | **71.8** | **92.1** | **85.9** | **74.7** |
| w/o HelpSteer3 | 90.4 | 83.5 | 68.0 | 91.5 | 84.4 | 70.3 |
| w/o Skywork | 90.6 | 84.2 | 66.1 | 91.3 | 85.1 | 69.7 |
| w/o Math-Step-DPO | 91.0 | 84.7 | 70.7 | 91.5 | 85.0 | 72.6 |
| w/o Code-Preference | **91.1** | 84.8 | 71.2 | 91.8 | 85.4 | 74.0 |

To understand each dataset's contribution, we conducted an ablation study, systematically removing one at a time to measure its impact on model performance. From Table 4, this analysis revealed a clear distinction between the roles of foundational and specialized datasets. For HelpSteer3, removing this foundational dataset led to a broad performance decline, with a 4.4-point drop on the RMB benchmark and a 1.5-point drop on RMBench. This highlights its critical role in establishing general understanding and factual accuracy. For Skywork, excluding our only dataset with safety data caused the most severe impact—a 5.0-point drop on the safety-focused RMB benchmark. This confirms that explicit safety training is essential for model harmlessness. For Math-Step-DPO, removing this specialized dataset predictably weakened reasoning, causing a 2.1-point drop on RMB and a 0.9-point drop on RMBench. This shows in-domain data is necessary for complex, multi-step reasoning. For Code-Preference, the exclusion of this smaller dataset resulted in the most modest decrease (0.7 points on RMB and 0.5 points on RMBench). This suggests its logical reasoning contributions are partially covered by the broader datasets.

## 5 CONCLUSION

We recast reward modeling as a focused, two-stage judgment rather than a single holistic score. Our reward model first identifies a small set of instance-critical hypotheses, then performs a targeted reread conditioned on those hypotheses – directing compute where risk is highest. This simple shift curbs judgment diffusion, reduces style-driven bias, and exposes subtle factual or logical slips, yielding more stable preferences across benchmarks. In essence, the same "think twice" discipline that improves solvers also makes their judges more reliable. Future work will focus on integrating verification tools like retrieval and code runners to check claims, automatically generating dynamic evaluation rules to replace static rubrics, and using model uncertainty to adaptively decide when and how deeply to re-evaluate an output.

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

**Content of Appendix**

# A PRELIMINARY STUDY ON JUDGMENT DIFFUSION

We design a preliminary study to quantify the focus of RMs on multiple evaluation dimensions by analyzing the token allocation of model generation to verify our concepts of focus dilution and shallow analysis.

Our experimental setup involves comparing two models head-to-head. The first, our baseline, RRM-32B (Guo et al., 2025), is a standard generative RM trained to produce a holistic critique in a single forward pass, thereby considering all evaluation criteria at once. The second is our proposed two-turn branch-and-rethink approach, which executes both the Adaptive Branching and Branch-Conditioned Rethinking steps. Crucially, both models are given the exact same maximum token limit, 16384. This ensures we're measuring how they choose to spend a fixed test-time compute budget. For the dataset, we selected two challenging subsets from the RM-Bench benchmark to test distinct capabilities. From the Chat_Hard split, we sample 200 instances manually that require sharp factual accuracy and solid logical reasoning to judge correctly. From the Code split, we sampled 200 instances where the judgment hinges on catching subtle bugs, checking algorithmic correctness, and validating the implementation.

We follow a clear procedure to gather our results. First, for every task in our test sets, we have both the Baseline and Our Method generate their respective judgments. The next and most critical step is **Token Allocation**. Here, we prompt a large language model, GPT-4o-0806, to read the generated text from each model and attribute every single sentence to one of the predefined evaluation dimensions (e.g., Information Accuracy, Logical Reasoning). For our method, we combine the tokens from both the initial Adaptive Branching turn and the final Branch-Conditioned Rethinking turn. After that, we calculate the proportions by determining the percentage of total test-time compute (tokens) spent on each dimension for every instance. Finally, we average these percentage allocations across all 200 instances within each split.

As shown in Figure 1, the baseline model exemplifies the problems of judgment diffusion. On both (a) Chat_Hard and (b) Code tasks, it distributes its tokens broadly across numerous dimensions, such as Writing Clarity and Content Relevance, even when they are not critical considering the strong power of current LLMs. This is a clear indicator of focus dilution, as the model's limited generative budget is spread thin instead of being concentrated on what matters. This, in turn, leads to a shallow analysis of the most important factors. For instance, in the Code task, the baseline allocates only about 25% of its tokens to Implementation Capability. In stark contrast, our method demonstrates adaptive focus: it concentrates over 70% of its generation on Information Accuracy and Logical Reasoning for Chat_Hard and over 75% on Implementation Capability and Computational Precision for Code. This evidence suggests that without a structured mechanism for focusing, RMs default to a diffuse and superficial evaluation, whereas our approach enables a targeted and deeper analysis.

## B EXPERIMENT SETTING

### B.1 DATASETS

We train on a diverse mixture of datasets to ensure both coverage and domain specialization:

- **HelpSteer3** (Wang et al., 2025): A large-scale, human-annotated dataset providing diverse open-domain preference pairs across multiple languages and tasks. Each pair is supplemented with explicit critiques, which makes it particularly well-suited for training generative and reasoning-based RMs. This dataset contributes the breadth needed to capture general human preferences in everyday conversational and instructional settings.

- **Skywork Reward Preference-80K** (Liu et al., 2024b): A broad-coverage dataset emphasizing instruction-following, reasoning quality, and safety compliance. Unlike HelpSteer3, which covers open-ended domains, Skywork's annotations focus on alignment with user intent and responsible behavior. This introduces strong supervision for safety- and adherence-related dimensions, which are often underrepresented in general instruction datasets.

- **Code-Preference-Pairs (8K)** (Vezora, 2024): A curated collection of programming-related preference pairs, highlighting tradeoffs between functional correctness, readability, and efficiency. This dataset provides focused supervision for *Implementation Capability* and *Computational Precision*, two dimensions frequently selected in adaptive branching for code tasks. Without such domain-specific data, reward models tend to overfit to stylistic features instead of true semantic correctness.

- **Math-Step-DPO-10K** (Lai et al., 2024): Preference pairs drawn from mathematical problem solving, where solutions may differ in intermediate reasoning steps as well as final correctness. This dataset is critical for training models to detect subtle reasoning errors, making it an ideal testbed for *Logical Reasoning* and *Information Accuracy*. It encourages the model not only to verify final answers but also to scrutinize the validity of intermediate steps.

### B.2 EVALUATION DATA

We evaluate our models on three widely used and challenging benchmarks that capture different facets of alignment quality. Together, they provide a comprehensive testbed spanning subtle reasoning, real-world coverage, and robustness to bias.

- **RewardBench** (Lambert et al., 2025b): A standard benchmark for evaluating reward models across diverse domains, including factual QA, multi-step reasoning, safety-critical instructions, and stylistic tradeoffs. It is widely adopted for measuring whether reward models align with human preferences in broad settings. RewardBench serves as our measure of *generalization breadth*: models must balance correctness, reasoning, safety, and fluency simultaneously.

- **RM-Bench** (Liu et al., 2025b): A more fine-grained evaluation that emphasizes sensitivity to *subtle errors* such as quiet factual slips, brittle logical steps, and superficial stylistic cues. RM-Bench is also designed to probe *systematic biases*, including verbosity bias (favoring longer responses) and position bias (favoring earlier/later candidates). Performance here reflects a model's ability to avoid shallow scanning and resist bias—precisely where our two-turn "branch-and-rethink" framework aims to excel.

- **RMB** (Zhou et al., 2025): A large-scale, comprehensive benchmark covering 49 real-world scenarios, ranging from open-domain dialogue and question answering to code, math, and safety. Unlike other benchmarks, RMB supports both pairwise preference comparisons and Best-of-N (BoN) evaluation, where models must select the best among multiple candidates. This setup better reflects practical deployment conditions, since real-world LLMs often generate diverse candidate sets. RMB explicitly reports performance on *helpfulness* and *harmlessness*, making it a balanced measure of alignment quality across both capability and safety.

### B.3 BASELINES

We compare against a wide array of state-of-the-art reward models, grouped by their architecture and output modality:

- **Scalar RMs:** Conventional reward models that directly output a single scalar score without generating intermediate reasoning. We evaluate widely used systems such as SteerLM-RM-70B, Eurus-RM-7B (Yuan et al., 2025), InternLM2-20B-Reward, InternLM2-7B-Reward (Cai et al., 2024), Skywork-Reward-Gemma-2-27B , Skywork-Reward-Llama-3.1-8B (Liu et al., 2024b), ArmoRM-Llama3-8B-v0.1, Nemotron-4-340B-Reward (Adler et al., 2024), and INFORM-Llama3.1-70B (Chen et al., 2025b).

- **Generative RMs (GenRMs):** Reward models that produce natural language critiques or rationales before collapsing them into a scalar score. This category includes both closed-source and open-source representatives: Claude-3.5-Sonnet (Anthropic, 2024), Llama-3.1-70B-Instruct (Dubey et al., 2024), Gemini-1.5-Pro (Reid et al., 2024), GPT-4o (OpenAI, 2023), and Skywork-Critic-Llama-3.1-70B (Liu et al., 2024b).

- **Reasoning RMs (ReasoningRMs):** Advanced reward models that explicitly perform intermediate reasoning or structured critique before producing a preference. This includes JudgeLRM (Chen et al., 2025a), DeepSeek-PairRM-27B, DeepSeek-GRM-27B-RFT, DeepSeek-GRM-27B (Liu et al., 2025c), Self-Taught-Evaluator-Llama3.1-70B (Wang et al., 2024), J1-Llama-70B (Whitehouse et al., 2025), EvalPlanner-Llama-3.3-70B-Instruct (Saha et al., 2025), as well as RM-R1 model family (Chen et al., 2025b).

We prioritize official results reported by model providers or benchmark organizers, or other published works (like Chen et al. (2025b)), and evaluate remaining models under their standardized setup. If a paper provides multiple models, we report the strong ones.

## C   PROMPTS USED IN BR-RM

Here we present the prompts implemented in our two-turn, branch-and-rethink reward modeling framework. The first-stage prompt in Figure 3, operationalizes the **Adaptive Branching** step. It instructs the reward model (RM) to act as a quality evaluator, compelling it to first select one to three critical cognitive dimensions from a predefined list in the `[Quality Assessment Focus]` section. This step forces the model to narrow its attention. Subsequently, the RM generates a structured issue analysis for each response, focusing only on the chosen dimensions. The second-stage prompt in Figure 4 implements **Branch-Conditioned Rethinking**. It consumes the structured output from the first stage and guides the model to apply a relevant evaluation hierarchy (e.g., `Correctness > Process > Presentation`) to the identified issues. This conditioned re-evaluation culminates in a final comparative score, `\boxed{1 or 2}`, which provides the clean, binary reward signal used for training.

---

**Turn 1: Adaptive Branching**

You are a **response quality evaluator**. Given the context of the conversation (the last turn is the User's query) and two responses from the Assistant, you should **compare the difference** of two model responses, **select the most important cognitive abilities** for this query, and **analyze critical issues** in each response.

**Context:**
{context}

**Responses:**
{responses}

**Output Format:**
[Quality Assessment Focus]
**Choose 1-3 abilities**: Information Accuracy, Computational Precision, Logical Reasoning, Implementation Capability, Safety Awareness, Response Completeness, Instruction Adherence, Communication Clarity.
[End of Quality Assessment Focus]

[Quality Analysis for Response 1]
- Critical Issues: [Focus on chosen abilities, list specific errors/concerns, or "None identified"]
  * Information Accuracy: factual errors, source reliability, misinformation
  * Computational Precision: calculation errors, formula mistakes, step validity
  * Logical Reasoning: conclusion correctness **(CRITICAL)**, logical flaws, reasoning gaps
  * Implementation Capability: functional errors **(CRITICAL)**, security issues, inefficiency
  * Safety Awareness: harmful content **(CRITICAL)**, inappropriate refusals, bias
  * Instruction Adherence: constraint violations, format errors, requirement misses
  * Response Completeness: missing content, insufficient detail, incomplete coverage
[End of Quality Analysis for Response 1]

[Quality Analysis for Response 2]
- Critical Issues: [Same format as above]
[End of Quality Analysis for Response 2]

---

Figure 3: Prompt for adaptive branching.

---

**Turn 2: Branch-Conditioned Rethinking**

You are making **final comparative judgments** using **established evaluation priorities**. You have the conversation context, two responses to compare, and a detailed quality analysis from a previous evaluation.
Before scoring, **analyze step by step**. **Different query types require different evaluation hierarchies**. Please **strictly follow** the required output format.

**Evaluation Hierarchies:**
- **Accuracy-Critical** (factual, computational, technical): **Correctness > Process > Presentation**
- **Creative/Open-Ended** (writing, discussion): **User Intent > Content Quality > Creativity**
- **Instruction-Following** (constrained tasks): **Adherence > Content > Clarity**

#### **Output Format Requirements** ####
[The Beginning of Analysis on Response 1]
[Apply appropriate evaluation hierarchy to the quality analysis findings]
[The End of Analysis on Response 1]

[The Beginning of Analysis on Response 2]
[Apply appropriate evaluation hierarchy to the quality analysis findings]
[The End of Analysis on Response 2]

[The Beginning of Ranking Score]
\boxed{1 or 2} (response that better meets the appropriate evaluation hierarchy)
[The End of Ranking Score]

---

Figure 4: Prompt for branch-conditioned rethinking.

# D    IMPLEMENTATION DETAILS

Table 5: Training and Inference Configurations.

| Category | Parameter | Value |
|---|---|---|
| **Distributed Training** | Total Nodes | 8 |
| | GPUs per Node | 8 |
| | Total GPUs | 64 |
| | Tensor Parallelism | 8 |
| | Context Parallelism | 1 |
| | Pipeline Parallelism | 1 |
| | Memory Optimization | FSDP2 (DTensor) |
| **Training Hyperparameters** | Optimization Algorithm | GRPO |
| | Training Steps | 400 |
| | Global Batch Size | 256 |
| | Micro Batch Size | 1 |
| | Prompts per Step | 128 |
| | Generations per Prompt | 8 |
| | Temperature | 1.0 |
| | PPO Clip Min | 0.2 |
| | PPO Clip Max | 0.28 |
| | KL Penalty | 0.001 |
| | Weight Decay | 0.0 |
| | Gradient Clip Norm | 1.0 |
| | Adam $\beta$ | 0.9 |
| | Adam $\epsilon$ | $1 \times 10^{-8}$ |
| | Learning Rate (8B) | $5 \times 10^{-7}$ |
| | Learning Rate (14B) | $1 \times 10^{-6}$ |
| | Max Total Tokens | 16384 |
| | Precision | bfloat16 |
| **Evaluation Hyperparameters** | Batch Size | 1,024 |
| **Generation Settings** | Backend | vLLM |
| | Top-p | 0.95 |
| | Top-k | 20 |
| | Temperature | 1.0 |
| | Max New Tokens | 8,192 |
| | Max Total Tokens | 16,384 tokens |

Our implementation leverages a scalable and efficient post-training library, NeMo-RL[2], with a custom two-turn reward modeling environment. Regarding model design, the penalty constant for format checking is -100. The used prompts are in Appendix C The training pipeline processes each stage sequentially, applying stage-specific stop strings and parsing structured outputs using regex-based extraction. We employ Group Relative Policy Optimization (GRPO, Shao et al. 2024), which extends PPO by optimizing over groups of responses simultaneously, providing more stable gradients for preference learning. The distributed training configuration utilizes 8 nodes with 8 GPUs each, employing fully sharded data parallel with tensor parallelism of 8 across GPUs. Memory optimization techniques include CPU offloading for optimizer states, activation checkpointing, and mixed precision training with bfloat16. The training process uses asymmetric GRPO clipping bounds [0.2, 0.28] for variance reduction and a KL divergence penalty of 0.001. Model-specific learning rates were tuned based on scale: $5 \times 10^{-7}$ for 8B models and $1 \times 10^{-6}$ for 14B models, all trained for 400 optimization steps with validation every 5 steps. For inference, we employ vLLM with async engine support for high-throughput evaluation. The evaluation protocol maintains consistency with training parameters, using temperature 1.0 for generation and computing accuracy as the primary metric, with additional domain-specific metrics for comprehensive analysis. More configuration details are shown in Table 5 in the Appendix. Table 5 presents the key hyperparameters used for training our models. These parameters were carefully selected to optimize the reinforcement learning process and ensure effective development of reasoning capabilities in our models.

---

[2]https://github.com/NVIDIA-NeMo/RL

# E  DETAILED EXPERIMENT RESULTS

In this section, we provide the full experiment results and a more comprehensive coverage of existing baselines. The results of RewardBench, RM-Bench, and RMB are provided in Tables 6, 7, and 8.

Table 6: Results on the RewardBench. * indicates potential data contamination.

| Model | Chat | Chat_Hard | Safety | Reasoning | Overall |
|---|---|---|---|---|---|
| **ScalarRMs** | | | | | |
| Eurus-RM-7b | 98.0 | 65.6 | 81.4 | 86.3 | 82.8 |
| Internlm2-7b-reward | 99.2 | 69.5 | 87.2 | 94.5 | 87.6 |
| SteerLM-RM-70B | 91.3 | 80.3 | 92.8 | 90.6 | 88.8 |
| Cohere-0514 | 96.4 | 71.3 | 92.3 | 97.7 | 89.4 |
| Internlm2-20b-reward | 98.9 | 76.5 | 89.5 | 95.8 | 90.2 |
| ArmorRM-Llama3-8B-v0.1 | 96.9 | 76.8 | 90.5 | 97.3 | 90.4 |
| Nemotron-4-340B-Reward | 95.8 | 87.1 | 91.5 | 93.6 | 92.0 |
| Skywork-Reward-Llama-3.1-8B* | 95.8 | 87.3 | 90.8 | 96.2 | 92.5 |
| Skywork-Reward-Gemma-2-27B* | 95.8 | 91.4 | 91.9 | 96.1 | 93.8 |
| INF-ORM-Llama3.1-70B | 96.6 | 91.0 | 93.6 | 99.1 | 95.1 |
| **GenRMs** | | | | | |
| Llama3.1-8B-Instruct | 85.5 | 48.5 | 75.6 | 72.1 | 70.4 |
| Prometheus-8*7B-v2 | 93.0 | 47.1 | 80.5 | 77.4 | 74.5 |
| Llama3.1-70B-Instruct | 97.2 | 70.2 | 82.8 | 86.0 | 84.0 |
| Llama3.1-405B-Instruct | 97.2 | 74.6 | 77.6 | 87.1 | 84.1 |
| Claude-3-5-sonnet-20240620 | 96.4 | 74.0 | 81.6 | 84.7 | 84.2 |
| GPT-4o-0806 | 96.1 | 76.1 | 86.6 | 88.1 | 86.7 |
| Gemini-1.5-pro | 92.3 | 80.6 | 87.9 | 92.0 | 88.2 |
| SFR-LlaMa-3.1-70B-Judge-r | 96.9 | 84.8 | 91.6 | 97.6 | 92.7 |
| Skywork-Critic-Llama-3.1-70B* | 96.6 | 87.9 | 93.1 | 95.5 | 93.3 |
| **ReasonRMs** | | | | | |
| JudgeLRM | 92.9 | 56.4 | 78.2 | 73.6 | 75.2 |
| SynRM | 38.0 | 82.5 | 74.1 | 87.1 | 70.4 |
| RM-R1-DeepSeek-DISTILLED-Qwen-7B | 88.9 | 66.2 | 78.4 | 87.0 | 80.1 |
| Cloud | 97.0 | 58.0 | 84.0 | 92.0 | 82.8 |
| DeepSeek-GRM-16B | 90.8 | 74.3 | 84.7 | 81.8 | 82.9 |
| DeepSeek-GRM-27B-RFT | 94.7 | 77.2 | 87.0 | 79.2 | 84.5 |
| RM-R1-Qwen-INSTRUCT-7B | 94.1 | 74.6 | 85.2 | 86.7 | 85.2 |
| DeepSeek-GRM-27B | 94.1 | 78.3 | 88.0 | 83.8 | 86.0 |
| DeepSeek-PairRM-27B | 95.3 | 86.8 | 52.3 | 92.0 | 87.1 |
| RM-R1-Qwen-INSTRUCT-14B | 93.6 | 80.5 | 86.9 | 92.0 | 88.2 |
| RM-R1-DeepSeek-DISTILLED-Qwen-14B | 91.3 | 79.4 | 89.3 | 95.5 | 88.9 |
| Self-taught-evaluator-Llama3.1-70B | 96.9 | 83.1 | 89.6 | 88.4 | 90.0 |
| EvalPlanner-Llama-3.3-70B-Instruct | 97.7 | 89.5 | 91.7 | 96.1 | 93.8 |
| RM-R1-DeepSeek-DISTILLED-Qwen-32B | 95.3 | 80.3 | 91.1 | 96.8 | 90.9 |
| RM-R1-Qwen-INSTRUCT-32B | 95.3 | 83.1 | 91.9 | 95.2 | 91.4 |
| **Our Models** | | | | | |
| BR-RM-Qwen-8B | 95.8 | 80.1 | 90.4 | 97.5 | 91.0 |
| BR-RM-Qwen-14B | 97.0 | 82.4 | 90.0 | 98.8 | 92.1 |

Table 7: The full results of tested reward models on RM-Bench. Chat, Math, Code, and Safety show the model's Average accuracy on each domain. Easy, Normal, Hard show the model's Accuracy on each difficulty level across all domains. **Bold** numbers indicate the best performance, Underlined numbers indicate the second best.

| Model | Chat | Math | Code | Safety | Easy | Normal | Hard | Avg |
|---|---|---|---|---|---|---|---|---|
| **ScalarRMs** | | | | | | | | |
| steerlm-70b | 56.4 | 53.0 | 49.3 | 51.2 | 48.3 | 54.9 | 54.3 | 52.5 |
| tulu-v2.5-70b-preference-mix-rm | 58.2 | 51.4 | 55.5 | 87.1 | 72.8 | 65.6 | 50.7 | 63.0 |
| Mistral-7B-instruct-Unified-Feedback | 56.5 | 58.0 | 51.7 | 86.8 | 87.1 | 67.3 | 35.3 | 63.2 |
| RM-Mistral-7B | 57.4 | 57.0 | 52.7 | 87.2 | 88.6 | 67.1 | 34.9 | 63.5 |
| Eurus-RM-7b | 59.9 | 60.2 | 56.9 | 86.5 | 87.2 | 70.2 | 40.2 | 65.9 |
| internlm2-7b-reward | 61.7 | 71.4 | 49.7 | 85.5 | 85.4 | 70.7 | 45.1 | 67.1 |
| Skywork-Reward-Gemma-2-27B | 69.5 | 54.7 | 53.2 | 91.9 | 78.0 | 69.2 | 54.9 | 67.3 |
| ArmorRM-Llama3-8B-v0.1 | 67.8 | 57.5 | 53.1 | 92.4 | 82.2 | 71.0 | 49.8 | 67.7 |
| GRM-llama3-8B-sftreg | 62.7 | 62.5 | 57.8 | 90.0 | 83.5 | 72.7 | 48.6 | 68.2 |
| internlm2-20b-reward | 63.1 | 66.8 | 56.7 | 86.5 | 82.6 | 71.6 | 50.7 | 68.3 |
| llama-3-OffsetBias-RM-8B | 71.3 | 61.9 | 53.2 | 89.6 | 84.6 | 72.2 | 50.2 | 69.0 |
| Nemotron-340B-Reward | 71.2 | 59.8 | 59.4 | 87.5 | 81.0 | 71.4 | 56.1 | 69.5 |
| URM-LLaMA-3.1-8B | 71.2 | 61.8 | 59.1 | 93.1 | 84.0 | 73.2 | 53.0 | 70.0 |
| Skywork-Reward-Llama-3.1-8B | 69.5 | 60.6 | 54.5 | **95.7** | 89.0 | 74.7 | 46.6 | 70.1 |
| INF-ORM-Llama3.1-70B | 66.3 | 65.6 | 56.8 | 94.8 | 91.8 | 76.1 | 44.8 | 70.9 |
| **GenRMs** | | | | | | | | |
| tulu-v2.5-dpo-13b-chatbot-arena-2023 | 64.9 | 52.3 | 50.5 | 62.3 | 82.8 | 60.2 | 29.5 | 57.5 |
| tulu-v2.5-dpo-13b-nectar-60k | 56.3 | 52.4 | 52.6 | 73.8 | 86.7 | 64.3 | 25.4 | 58.8 |
| stablelm-2-12b-chat | 67.2 | 54.9 | 51.6 | 65.2 | 69.1 | 63.5 | 46.6 | 59.7 |
| tulu-v2.5-dpo-13b-stackexchange-60k | 66.4 | 49.9 | 54.2 | 69.0 | 79.5 | 63.0 | 37.2 | 59.9 |
| Nous-Hermes-2-Mistral-7B-DPO | 58.8 | 55.6 | 51.3 | 73.9 | 69.5 | 61.1 | 49.1 | 59.9 |
| Claude-3-5-sonnet-20240620 | 62.5 | 62.6 | 54.4 | 64.4 | 73.8 | 63.4 | 45.9 | 61.0 |
| tulu-v2.5-dpo-13b-hh-rlhf-60k | 68.4 | 51.1 | 52.3 | 76.5 | 53.6 | 63.0 | 69.6 | 62.1 |
| tulu-2-dpo-13b | 66.4 | 51.4 | 57.8 | 85.4 | 86.9 | 66.7 | 37.7 | 63.8 |
| SOLAR-10.7B-Instruct-v1.0 | **78.6** | 52.3 | 49.6 | 78.9 | 57.5 | 67.6 | 69.4 | 64.8 |
| Llama3.1-70B-Instruct | 64.3 | 67.3 | 47.3 | 83.0 | 74.7 | 67.8 | 54.1 | 65.5 |
| Skywork-Critic-Llama-3.1-70B | 71.4 | 64.0 | 56.8 | 94.8 | 85.6 | 73.7 | 56.5 | 71.9 |
| GPT-4o-0806 | 67.2 | 67.5 | 63.6 | 91.7 | 83.4 | 75.6 | 58.7 | 72.5 |
| Gemini-1.5-pro | 71.6 | 73.9 | 63.7 | 91.3 | 83.1 | 77.6 | 64.7 | 75.2 |
| **ReasonRMs** | | | | | | | | |
| JudgeLRM | 59.9 | 59.9 | 51.9 | 87.3 | 73.2 | 66.2 | 54.8 | 64.7 |
| RM-R1-Qwen-INSTRUCT-7B | 66.6 | 67.0 | 54.6 | 92.6 | 79.2 | 71.7 | 59.7 | 70.2 |
| Self-taught-evaluator-llama3.1-70B | 73.4 | 65.7 | 56.3 | 90.4 | 80.2 | 74.5 | 59.7 | 71.5 |
| EvalPlanner-Llama-3.3-70B-Instruct | – | – | – | – | 81.1 | 80.8 | **84.3** | 82.1 |
| RM-R1-DeepSeek-DISTILLED-Qwen-7B | 64.0 | 83.9 | 56.2 | 85.3 | 75.9 | 73.1 | 68.1 | 72.4 |
| RM-R1-Qwen-INSTRUCT-14B | 75.6 | 75.4 | 60.6 | 93.6 | 82.6 | 77.5 | 68.8 | 76.1 |
| RM-R1-Qwen-INSTRUCT-32B | 75.3 | 80.2 | 66.8 | 93.9 | 86.3 | 80.5 | 70.4 | 79.1 |
| RM-R1-DeepSeek-DISTILLED-Qwen-32B | 71.8 | 90.5 | 69.5 | 94.1 | 86.2 | 83.6 | 74.4 | 81.5 |
| RM-R1-DeepSeek-DISTILLED-Qwen-14B | 74.2 | 91.8 | 74.1 | 95.4 | 89.5 | 85.4 | 76.7 | 83.9 |
| **Our Models** | | | | | | | | |
| BR-RM-Qwen-8B | 76.4 | **94.1** | 77.0 | 92.7 | 91.7 | 87.3 | 76.1 | 85.0 |
| BR-RM-Qwen-14B | 77.3 | 92.6 | **79.8** | 93.7 | **92.0** | **88.1** | 77.6 | **86.1** |

Table 8: The leaderboard of RMB, ranked by the average score of all subsets. **Bold** numbers indicate the best performance, Underlined numbers indicate the second best.

| Model | Helpfulness | | Harmlessness | | |
| --- | --- | --- | --- | --- | --- |
| | BoN | Pairwise | BoN | Pairwise | Overall |
| **ScalarRMs** | | | | | |
| Tulu-v2.5-13b-preference-mix-rm | 0.355 | 0.562 | 0.351 | 0.545 | 0.453 |
| SteerLM-RM 70B | 0.502 | 0.574 | 0.578 | 0.673 | 0.582 |
| Skywork-Reward-Gemma-2-27B | 0.472 | 0.653 | 0.561 | 0.721 | 0.602 |
| Internlm2-20b-reward | 0.585 | 0.763 | 0.499 | 0.670 | 0.629 |
| ArmorRM-Llama3-8B-v0.1 | 0.636 | 0.787 | 0.497 | 0.663 | 0.646 |
| Internlm2-7b-reward | 0.626 | 0.782 | 0.563 | 0.712 | 0.671 |
| Eurus-RM-7b | 0.679 | 0.818 | 0.543 | 0.693 | 0.683 |
| Skywork-Reward-Llama-3.1-8B | 0.627 | 0.781 | 0.603 | 0.759 | 0.693 |
| INF-ORM-Llama3.1-70B | 0.650 | 0.798 | 0.607 | 0.767 | 0.705 |
| Starling-RM-34B | 0.604 | 0.774 | 0.674 | 0.795 | 0.712 |
| **GenRMs** | | | | | |
| Llama2-70b-chat | 0.289 | 0.613 | 0.249 | 0.602 | 0.438 |
| Llama3.1-8B-Instruct | 0.365 | 0.675 | 0.267 | 0.653 | 0.490 |
| Gemini-1.5-pro | 0.536 | 0.763 | 0.299 | 0.661 | 0.565 |
| Mixtral-8x7B-Instruct-v0.1 | 0.480 | 0.706 | 0.491 | 0.671 | 0.587 |
| skywork-critic-llama3.1-8B | 0.600 | 0.725 | 0.578 | 0.578 | 0.620 |
| skywork-critic-llama3.1-70B | 0.640 | 0.753 | 0.614 | 0.614 | 0.655 |
| Llama3.1-70B-Instruct | 0.648 | 0.811 | 0.558 | 0.739 | 0.689 |
| Mistral-Large-2407 | 0.678 | 0.817 | 0.583 | 0.725 | 0.701 |
| Claude-3-5-sonnet | **0.705** | **0.838** | 0.518 | 0.764 | 0.706 |
| Qwen2-72B-Instruct | 0.645 | 0.810 | 0.649 | 0.789 | 0.723 |
| GPT-4o-2024-05-13 | 0.639 | 0.815 | 0.682 | 0.814 | 0.738 |
| **ReasonRMs** | | | | | |
| JudgeLRM | 0.363 | 0.699 | 0.363 | 0.674 | 0.531 |
| RM-R1-DeepSeek-DISTILLED-Qwen-7B | 0.451 | 0.658 | 0.429 | 0.664 | 0.551 |
| RM-R1-Qwen-INSTRUCT-7B | 0.543 | 0.740 | 0.608 | 0.765 | 0.664 |
| Self-taught-evaluator-llama3.1-70B | 0.616 | 0.786 | 0.546 | 0.733 | 0.670 |
| Deepseek-27B-RFT | 0.592 | 0.801 | 0.548 | 0.765 | 0.670 |
| RM-R1-DeepSeek-DISTILLED-Qwen-14B | 0.593 | 0.765 | 0.613 | 0.769 | 0.685 |
| Deepseek-GRM-27B | 0.623 | 0.805 | 0.570 | 0.761 | 0.690 |
| RM-R1-Qwen-INSTRUCT-14B | 0.594 | 0.776 | 0.620 | 0.778 | 0.692 |
| RM-R1-DeepSeek-DISTILLED-Qwen-32B | 0.620 | 0.782 | 0.618 | 0.771 | 0.698 |
| RM-R1-Qwen-INSTRUCT-32B | 0.636 | 0.791 | 0.682 | 0.809 | 0.730 |
| **Our Models** | | | | | |
| BR-RM-Qwen-8B | 0.597 | 0.769 | 0.685 | **0.822** | 0.718 |
| BR-RM-Qwen-14B | 0.670 | 0.810 | **0.693** | 0.816 | **0.747** |

## F  CASE STUDY

To demonstrate the effectiveness of our Branch-and-Rethink Reasoning Reward Model, we present a case study involving the evaluation of C++ function implementations. The user query requests a function `unique_digits(vector<int> x)` that returns a sorted vector containing only integers without any even digits. This programming task represents a typical scenario where multiple correct implementations may exist with subtle quality differences that traditional reward models might overlook.

**Turn 1: Adaptive Branching**  In the first turn, our model performs adaptive branching by dynamically selecting the most relevant evaluation dimensions for this specific task. Rather than applying a fixed set of criteria, the model identifies three critical dimensions based on the task context:

1. **Implementation Capability** – Assessing the correctness of the algorithmic logic and code structure

2. **Computational Precision** – Evaluating the accuracy of digit-checking operations and numerical computations

3. **Instruction Adherence** – Verifying compliance with specified requirements, including sorting and filtering constraints

During this phase, the model conducts focused analysis on both responses. For Response 1, the model identifies a properly structured helper function for digit checking, correct filtering logic, and the inclusion of a complete `main` function with example usage. Response 2 exhibits functionally identical core logic but presents a commented-out `main` function. Notably, both responses pass the initial screening with no critical errors identified, a scenario that challenges traditional single-pass evaluation methods.

**Turn 2: Branch-Conditioned Rethinking**  The second turn leverages the insights from adaptive branching to conduct deeper, issue-driven analysis. Based on the absence of critical errors identified in Turn 1, the model applies a hierarchical evaluation framework prioritizing *Correctness > Process > Presentation*. This accuracy-critical hierarchy ensures that functional correctness remains paramount while still distinguishing between responses of similar quality.

The branch-conditioned rethinking reveals subtle but meaningful differences:

- **Response 1** provides a fully functional implementation with an active `main` function and verifiable example cases, demonstrating completeness in both solution and presentation

- **Response 2** offers equivalent algorithmic correctness but with marginally reduced completeness due to the commented-out main function, which may impact immediate usability

Through this two-turn process, the model generates a nuanced ranking (Response 1 → Rank 1) that captures quality distinctions beyond binary correctness. This granular evaluation is crucial for reinforcement learning applications where the reward signal must differentiate between multiple acceptable solutions to guide model improvement effectively.

## G  STATEMENT ON USAGE OF LARGE LANGUAGE MODELS

In the preparation of this submission, large language models were utilized as a writing assistance tool. The primary role of the LLM was to enhance the clarity, grammar, and readability of the text. Its use was confined to language polishing, such as correcting syntax and improving sentence structure. The LLM was not used for any core research activities, including ideation, data analysis, or the formulation of conclusions. All intellectual contributions, including the research questions, methodology, and scientific insights, are the original work of the human authors. We have meticulously reviewed and edited all text and assume full responsibility for the final content of this paper.

## Case Study: C++ Function Implementation

**User Query:**

"Write a C++ function `vector<int> unique_digits(vector<int> x)` to solve the following problem:

Given a vector of positive integers x, return a sorted vector of all elements that hasn't any even digit.

Note: Returned vector should be sorted in increasing order."

**Turn 1: Adaptive Branching Results**

[ Implementation Capability ]  [ Computational Precision ]  [ Instruction Adherence ]

**Response 1 Analysis:**

- Implementation: Correct helper function and main logic structure
- Precision: Accurate digit-checking with no calculation errors
- Adherence: Full compliance with sorting and filtering requirements
- **Critical Issues: None identified**

**Response 2 Analysis:**

- Implementation: Functionally identical to Response 1
- Precision: Accurate digit-checking logic
- Adherence: Full compliance, commented main function
- **Critical Issues: None identified**

**Turn 2: Branch-Conditioned Rethinking**

**Hierarchical Evaluation (Correctness > Process > Presentation)**

- **Response 1:** Fully functional with verifiable main function and examples
- **Response 2:** Identical logic but less complete presentation
- Both responses achieve correctness, Response 1 excels in presentation

Figure 5: Case Study on C++ Function Implementations. The framework operates in two sequential turns: (a) Turn 1: Adaptive Branching, where the model dynamically selects critical evaluation dimensions (Implementation Capability, Computational Precision, and Instruction Adherence) based on the user query context and performs focused analysis to detect specific issues; (b) Turn 2: Branch-Conditioned Rethinking, where the model conducts deeper, issue-driven re-evaluation using a hierarchical assessment framework (Correctness > Process > Presentation) to generate final ranking scores. The case study demonstrates the evaluation of two functionally similar responses to a unique_digits() implementation task, showing how the model's two-turn approach enables nuanced quality assessment despite both responses having no critical errors. The final reward judgment (Response 1 ranked higher) is determined by subtle presentation and completeness differences identified through the branch-conditioned rethinking process, illustrating the model's capability to provide fine-grained distinctions necessary for effective reinforcement learning optimization.

