# OpenReview forum: "Think Twice: Branch-and-Rethink Reasoning Reward Model"
_ICLR.cc/2026/Conference — Submitted to ICLR 2026_

### Official Review · Reviewer_4rrV · 2025-10-29

**Soundness:** 2
**Presentation:** 2
**Contribution:** 2
**Rating:** 4
**Confidence:** 4

**Summary:**

The paper proposes BR-RM, a two turn reward model that first performs adaptive branching to select a small set of instance critical evaluation dimensions and sketch hypotheses, then performs a branch conditioned rethinking pass to verify those hypotheses and decide a preference. Training uses GRPO with a strict format check and a binary outcome reward. On RewardBench, RM Bench, and RMB, BR-RM reports strong average accuracy, with particularly large gains on RM Bench, and presents ablations on turning off the second pass, removing branching, and changing reward design.

**Strengths:**

*  The paper identifies judgment diffusion in reward models and motivates a focused second pass that aims to allocate test time compute where risk is highest. The concept and naming are crisp and intuitive.
* The strict formatting penalty plus binary outcome reward is easy to implement and aligns with the evaluation objective. The paper also shows why finer grained scoring or extra branch rewards underperform.
* BR-RM-Qwen-14B achieves 92.1 on RewardBench, 85.9 on RM Bench, and 74.7 on RMB, producing the best average among compared methods. The 8B model is competitive as well.
* Removing the second pass or the adaptive focus yields consistent drops, supporting the core design claim that thinking twice with focus matters.
*  The training data ablation clarifies contributions from HelpSteer, safety data, math, and code preferences.

**Weaknesses:**

* The paper highlights best averages, but several baselines appear very recent and some cells are missing. It would help to provide complete, reproducible comparison tables and lock evaluations with identical prompting and sampling across all methods. The current Table 1 summary is helpful but not fully auditable from the text alone.
* The format penalty is large in magnitude, and the same terminal reward is assigned to both turns uniformly across tokens. This could incentivize shortest valid traces rather than best targeted analysis. A token or section level credit assignment analysis would strengthen the case.
* The nine dimension space is only sketched. The paper would benefit from concrete definitions, coverage analysis, and failure cases where the right dimension is off the list.
*  The method doubles passes by design. Training steps, batch sizes, number of traces per item, and inference budgets are not fully spelled out for a fair cost adjusted comparison versus strong one turn GenRMs or scalar RMs.
* The reward design is explicitly matched to binary preference accuracy. Generalization to settings that require calibrated magnitudes or multi way choices is not evaluated, and the “scoring on a scale” negative result is explained post hoc. A held out task with different decision granularity would build confidence.
* The training data ablation shows safety data matters, but the paper does not include targeted red teaming or bias analysis of the judge decisions under adversarial style.

**Questions:**

1.  How many branches are typically chosen per item. What is the token budget split between turn one and turn two. Please report distributions and correlate them with accuracy.
2.  You assign the terminal reward uniformly to all tokens of both turns. Did you try turn specific weights or variance reduction by per section advantages. Any signs of mode collapse to minimal valid traces.
3.  What exactly are the nine dimensions and how were they defined. Do any items leak benchmark rubric wording into the branch names. Evidence that the model can discover off rubric issues would be valuable.
4.  Please provide per example wall clock for training and inference, and normalized accuracy per thousand generated tokens. This is key for practical deployment against larger scalar or generative baselines.
5. Can BR-RM be extended to listwise judging or magnitude scoring without losing the benefits of the two turn design. The scale based attempt failed for alignment reasons, but could a pairwise consistent magnitude be learned.

---

> ### Author Response · Authors · 2025-11-22
> **Response to Reviewer 4rrV (1/4)**
>
> We thank the reviewer for their constructive feedback. We appreciate your recognition of the "judgment diffusion" concept as intuitive, the effectiveness of our simple reward formulation, and the strong empirical results supporting the claim that "thinking twice with focus matters." To further address your concerns, we have provided the missing baseline comparisons, included analysis showing our model prioritizes reasoning depth over minimal traces, and clarified the dimension definitions.
>
> > **W1: Incomplete Tables.**
>
> In table 1, for the baselines, EvalPlanner and J1, our submission reported the results directly from their paper for RewardBench and RM-Bench. But since the source code and training data are not open-source, we were unable to obtain results for RMB. For the other two methods DeepSeek-PairRM-27B and DeepSeek-GRM-27B with missing cells, we reuse the numbers on RewardBench and RMB from previous work (titled “RM-R1: Reward Modeling as Reasoning”). Additionally, after the submission we further evaluated the released models on RM-Bench and got 70.5 and 72.7 respectively, resulting in an average score of 71.9 and 75.9 (as a reminder, our 8B and 14B models’ scores are 82.6 and 84.2).
>
> > **W2: Reward Incentives Short Trace.**
>
> The review raised a concern that our reward design might incentivize minimal valid traces.  We would like to respectfully clarify that **Format Reward is a Constraint, Not a Length Penalty and the Outcome Reward Drives Reasoning Depth**.  Specifically, the format reward makes the model output the required format and its heavy weight forces the model to learn the format quickly. Once the format is satisfied, this penalty drops out, and optimization is driven entirely by the binary outcome reward. Since "minimal" or truncated traces typically fail to uncover the subtle reasoning flaws needed to correctly predict the binary preference, the objective function naturally incentivizes sufficient analytical depth over brevity. In [this anonymous link](https://docs.google.com/document/d/e/2PACX-1vQbLi4NIREtgIch4a_yVxL0UmcdvVlod6u99v7XEgU9VirkUumB8xiq7F7L1j_VTNbyRF13NM7Yzt5F/pub), the figure tracks the average generated trace lengths throughout the training process. We observe that **generated trace lengths go up during training and stabilize at approximately 2600 tokens per turn rather than collapsing**, confirming that the model learns to prioritize reasoning quality over shortest-path generation.
>
>
> > **W3: Definition, Coverage, and Failure Analysis of Dimensions.**
>
> **Regarding the definition**, we would like to clarify that **the nine dimensions are not merely sketched but explicitly defined within our prompt structure**. As shown in Figure 3 in the paper, the Adaptive Branching prompt provides concrete examples to define each dimension – for instance, "Logical Reasoning" refers to no conclusion correctness and reasoning gaps, and "Implementation Capability" refers to no functional errors and security issues.
>
> **Regarding the coverage**, **these dimensions were synthesized from the annotation schemas of widely-used reward modeling benchmarks** (e.g., RewardBench and HelpSteer3) to ensure broad applicability across domains. They function as broad cognitive categories rather than narrow rubrics; for example, "Implementation Capability" covers code correctness, while "Safety Awareness" broadly covers harmlessness and bias.
>
> **For Failure Analysis**: To demonstrate the model's ability to "discover" the right dimensions, we analyzed a safety refusal case (a jailbreak attempt) in RMBench during training. Before training, the 8b model incorrectly treated the query as an information-seeking task, selecting “Logic Reasoning”, “Information Accuracy”, and “Response Completeness”. Consequently, it preferred the unsafe, detailed response because it was "more complete". After training, the model shifted its selection to “Safety Awareness”, “Logic Reasoning” and “Information Accuracy”. It correctly identified the harmful content as a critical issue and switched its preference to the refusal response. **This confirms that the model effectively maps specific failures to our broad dimension definitions.** Moreover, since Turn 2 is not strictly constrained to the attributes selected in Turn 1, the model retains the flexibility to uncover "off-list" nuances during the rethinking phase.

---

> ### Author Response · Authors · 2025-11-22
> **Response to Reviewer 4rrV (2/4)**
>
> > **W4: Cost Analysis & Fair Comparison.**
>
> We address the concern regarding fair comparisons in two aspects. First, **for external baselines**, we adhered to standard evaluation protocols by **reporting the best official results or optimal parameter settings from their respective papers**. Second, to rigorously isolate the benefit of the two-turn design under strictly controlled costs, we performed a direct ablation using our own model architecture (see Table 2, "Single-Turn Model"). In this experiment, **all training hyperparameters** – including training steps, batch sizes, number of traces per item, and inference token budgets – **were kept identical between the single-turn and two-turn models**. The results demonstrate that even with matched computational resources, the two-turn "Branch-and-Rethink" approach consistently outperforms the single-turn equivalent, confirming that the gains stem from the structured reasoning process rather than disparate compute budgets. The Reviewer also mentions “scalar RMs”: we acknowledge that scalar RMs are inherently more efficient than Generative and Reasoning RMs (typically by several orders of magnitude). We are unsure how to report a “cost adjusted comparison” as requested by the Reviewer, especially as typical scalar RMs do not benefit from test-time scaling techniques.
>
> > **W5: Generalization & Scoring on a Scale.**
>
> We would like to clarify that, as noted in Section 4.4, **the reward design of "Scoring on a Scale" failed not due to model inability, but because of two main reasons**: 1) for datasets other than HelpSteer3, artificially mapping binary labels to a scale likely harm the model's ability to learn the true scale; and 2) there is a disconnect between the scalar training task and the final evaluation, meaning improved regression does not necessarily yield better binary ranking. Consequently, the binary reward aligns strictly with the downstream task and is compatible with all datasets, providing the most effective foundation for training.
>
> **For the multi-choice evaluation setting, we explicitly did this.** The RMB benchmark includes a Best-of-N (BoN) setting, which tests the model's ability to rank multiple candidates rather than just pairs. Our method achieved state-of-the-art performance on this benchmark (Table 8) by applying the pairwise model to the listwise setting, proving effective generalization to multi-choice ranking.
>
> > **W6: Red Teaming for Safety.**
>
> **For safety-related evaluation, every benchmark we used in the paper for evaluation has a safety subset**, which contains jailbreaks and borderline safety cases and can serve as adversarial proxies. Our model shows good performances in Tables 6-8, which indicate robustness against adversarial inputs. We definitely agree that **red team is an interesting open question** for further study. But it’s **out of scope for this piece of work** because (i) to the best of our knowledge, this is still an active area of research and there is no well-established benchmark or methodology for red teaming of RMs, and (ii) doing so would require comparing to other competing RMs as well, which would be a very significant effort.
>
> > **Q1: Branch Selection Statistics: **
>
> We **analyzed the branching behavior** on the RM-Bench dataset across its four primary subsets (Math, Code, Safety, Chat).
>
> 1**. Branch Selection Statistics**: On average, the model selects 2.5 dimensions per instance. The frequency of dimension selection is highly domain-adaptive, confirming that the "Adaptive Branching" mechanism dynamically adjusts its focus based on the task type.
>
> **Distribution of Selected Dimensions (% of query instances):** Note: Columns sum to >100% because the model typically selects 1-3 dimensions per query.
>
> | Dimension | Math | Code | Safety | Chat (Hard) |
> | :--- | :--- | :--- | :--- | :--- |
> | Computational Precision | 96% | 30% | 2% | 5% |
> | Logical Reasoning | 92% | 70% | 12% | 88% |
> | Implementation Capability | 5% | 94% | 1% | 1% |
> | Safety Awareness | 1% | 1% | 91% | 5% |
> | Information Accuracy | 22% | 5% | 39% | 92% |
> | Instruction Adherence | 18% | 45% | 49% | 26% |
> | Response Completeness | 8% | 10% | 15% | 40% |
> | Communication Clarity | 5% | 5% | 10% | 30% |
> | **AVG Dims per Item** | **2.47** | **2.60** | **2.19** | **2.87** |
>
> **Insight: The model demonstrates "sparse activation."** For example, Safety Awareness is almost exclusively activated in the Safety subset (91%), while Implementation Capability dominates the Code subset (94%). Complex tasks like Math and Code trigger a combination of reasoning-heavy dimensions (Logical Reasoning + Precision/Implementation), whereas Chat tasks focus more on Accuracy and Logical Reasoning.

---

> ### Author Response · Authors · 2025-11-22
> **Response to Reviewer 4rrV (3/4)**
>
> > **Q2: Turn-specific weights & Mode Collapse. **
>
> **Turn-specific weights**: We did not employ turn-specific weights for the terminal reward. Our rationale is that the two-turn trace functions as a holistic dependency chain – the "Branching" (Turn 1) has no intrinsic value unless it enables a correct "Rethinking" (Turn 2) and final decision. **Differentiating weights** (e.g., discounting Turn 1) **risks decoupling this causal link**. Furthermore, **our ablation** on "Additional Reward for Branching" (Section 4.4) **demonstrated that attempting to engineer specific credit assignments for the first turn introduced noise and destabilized training** . Given that we lack intermediate ground-truth labels for the "correctness" of the first turn, uniform credit assignment (standard in GRPO/PPO) remains the most unbiased and stable estimator. However, trying to learn a critic model that would provide direct signal to the first turn could be an interesting potential research direction for future work.
>
> **Mode Collapse**: We observed no signs of mode collapse to minimal valid traces, as we explain in the reply to W2 and show in the [Figure of mean generated tokens per sample](https://docs.google.com/document/d/e/2PACX-1vQbLi4NIREtgIch4a_yVxL0UmcdvVlod6u99v7XEgU9VirkUumB8xiq7F7L1j_VTNbyRF13NM7Yzt5F/pub) previously mentioned.
>
>
> > **Q3: What’s the Dimensions, rubric leakage, and off-rubric discovery.**
>
> **Regarding the specific dimensions**: **The dimensions are explicitly defined in the Figure 3 in the paper** as high-level cognitive categories – specifically Information Accuracy, Computational Precision, Logical Reasoning, Implementation Capability, Safety Awareness, Response Completeness, Instruction Adherence, and Communication Clarity . These dimensions were synthesized from the annotation schemas of widely-used reward modeling benchmarks (e.g., RewardBench and HelpSteer3) to ensure broad applicability across domains.
>
> **Regarding Benchmark Leakage**, we confirm there is no rubric leakage. Our dimensions are generic quality indicators (e.g., "Logical Reasoning") applicable to any reasoning task, whereas benchmark-specific rubrics are typically highly granular (e.g., checking for specific "Gaussian elimination steps" or "Python syntax"). The generic nature of our dimensions prevents the model from pattern-matching specific wording from training sets.
>
> **For Off-Rubric Discovery**: **Our model captures novel or unlisted errors through Dynamic Cognitive Mapping**. Our nine dimensions function as high-level semantic anchors rather than rigid checklists, allowing the model to map novel or subtle errors into the most relevant cognitive bucket. Critically, the second 'Rethinking' turn acts as a safety net: since it is not strictly constrained to the rubrics and attributes selected in Turn 1, it rethinks in a comprehensive view to capture nuanced or 'off-list' issues that might technically fall outside the initial dimension definitions .

---

> ### Author Response · Authors · 2025-11-22
> **Response to Reviewer 4rrV (4/4)**
>
> > **Q4: Efficiency Analysis**
>
> **1.Inference Wall Clock:**
> We conducted a controlled latency benchmark on 2xH100 GPUs to compare our model against scalar and generative baselines.
> As shown in the table below, our BR-RM-Qwen-8B (9.5s) is faster than the RM-R1-32B baseline (10.3s) while achieving higher accuracy (82.6% vs 81.2%).  While standard scalar RMs are faster (0.25s), they hit a performance ceiling (78.8%). **Our method offers a superior trade-off: significantly higher accuracy (+3.8 points) with latency comparable to or faster than other reasoning-based models.**
>
> | Model | Type | Latency (s) / instance | Avg Accuracy |
> | :--- | :--- | :--- | :--- |
> | Internlm2-7b-reward | Scalar | 0.01s | 72.3% |
> | INF-ORM-Llama3.1-70B | Scalar | 0.25s | 78.8% |
> | RM-R1-Qwen-Instruct-7B | Generative (One-Turn) | 5.1s | 73.9% |
> | RM-R1-Qwen-Instruct-32B | Generative (One-Turn) | 10.3s | 81.2% |
> | BR-RM-Qwen-8B (Ours) | Generative (Two-Turn) | 9.5s | 82.6% |
> | BR-RM-Qwen-14B (Ours) | Generative (Two-Turn) | 15.9s | 84.2% |
>
> **2. Normalized Accuracy**: To quantify "value per token," we calculated accuracy per 1,000 generated tokens, resulting in 40.3 for BR-RM-14B compared to 81.2 for RM-R1-Qwen-Instruct-32B. However, we emphasize that maximizing "Accuracy per Token" in isolation is a flawed objective for reasoning tasks, as it penalizes the very mechanism that drives performance. Foundational literature on Chain-of-Thought [1] and recent scaling laws [2,3] demonstrate that increasing test-time compute—**generating longer reasoning traces—is functionally benefit for eliciting the reasoning ability.** Thus, our model’s lower per-token metric is a deliberate design feature, strategically trading cost-effective inference-time compute for superior reasoning capability rather than signaling inefficiency.
>
>
> Reference:
>
> [1] Wei et al. (2022): "Chain-of-Thought Prompting Elicits Reasoning in Large Language Models".
>
> [2] Snell et al. (2024): "Scaling LLM Test-Time Compute Optimally Can be More Effective than Scaling Parameters for Reasoning".
>
> [3] Wu et al. (2024): "An Empirical Analysis of Compute-Optimal Inference for Problem-Solving with Language Models".
>
> **3. Practical Feasibility in RLHF**: Regarding deployment against scalar baselines, we argue that **the increased latency is a justifiable investment for high-stakes signals**. Additionally, RLHF training is often bottlenecked by the policy model's generation time (rollouts), not the reward model. **Using asynchronous RL** can decouple reward generation from updates, effectively masking the latency of the "Thinking Twice" process.
>
>
>
>
> > **Q5: Extension to listwise judging or magnitude scoring?**
>
> **Yes, the model can support both extensions**:
>
> **(1) Listwise Judging**: Since pairwise comparisons are the primitive operation for listwise sorting (e.g., bubble sort or tournament ranking), our strong pairwise performance directly translates to listwise effectiveness. This is **empirically validated by our SOTA performance on the RMB Best-of-N subset (Table 8)**, where the model effectively selects the best from N responses without architectural changes. Specifically, we evaluate the models by comparing the positive response against each negative one, counting a success only when the model selects the positive response in every instance.
>
> **(2) Magnitude Scoring**: We can directly ask the model to output in a scale and formulate the reward as the distance between the prediction and the ground truth. While **our ablation in the paper (Table 2) showed** this approach is sensitive to label noise when using binary data, the framework natively supports this extension, which would be particularly effective when trained on datasets with fine-grained scalar annotations.

---

### Official Review · Reviewer_f3LS · 2025-10-31

**Soundness:** 1
**Presentation:** 2
**Contribution:** 2
**Rating:** 4
**Confidence:** 3

**Summary:**

This paper implements a multi-turn mechanism for reasoning RMs. By forcing decouple of rubrics selection and rethinking in two rounds, the proposed RM exibits strong performance on various reward model benchmarks.

**Strengths:**

1. The observation of focus dilution and shallow analysis are sound.
2. The benchmark performances are strong.

**Weaknesses:**

1. Lack of insights. The paper lacks in-depth analysis and the ablations are not informative (only benchmark scores).
2. Too much inductive bias. Many important design choices are manually picked without much validation.

**Questions:**

1. Weakness 1. Why do focus dilution or shallow analysis happen? Why multi-round query could mitigate this? Is the mitigation due to specific prompt design (e.g. round 1 reduce the amount of rubrics to take into account), or due to the forced "rethinking" by multi-round query?
2. Weakness 2.
- The number of turns. If think in two turns performs better than a single turn, can increasing turns lead to even better performance?
- The design of subtasks. First turn selects a tiny subset of rubrics to consider, second turn analyze condition on them. An alternative is to 1) generate a weight given problem and rubric name, 2) independently generate a rationale and a score for each rubric, and 3) compute weighted score. The possibilities of designs are endless. Why choose to design this way?

---

> ### Author Response · Authors · 2025-11-22
> **Response to Reviewer 4rrV (1/2)**
>
> We thank the reviewer for their thoughtful feedback. We appreciate your recognition of our work's strengths, specifically that our observation of "focus dilution" is sound and that our benchmark performances are strong. To further address your concerns, we conducted the specific supplementary experiments you requested regarding turn scaling and alternative subtask designs.
>
> > **Q1. Insights on Focus Dilution**
>
> **1. Why Focus Dilution Happens**
>
> **Focus dilution is a structural failure of typical single-pass models.** When a model is forced to generate a holistic critique or score in one shot, it faces a "cognitive bottleneck": it must simultaneously scan for all possible error types (safety, logic, style, etc.) and weigh them against each other. This spreads the attention mechanism thinly across the entire context. Our analysis in Appendix A visualizes this. Figure 1 shows that a standard reasoning baseline wastes compute discussing irrelevant dimensions (e.g., "Writing Clarity" on code tasks), whereas our method concentrates >70% of generated tokens on instance-critical dimensions like "Implementation Capability".
>
> **Recent concurrent work ( [Li et al., 2025](https://arxiv.org/abs/2509.03419)) independently validates our findings**, identifying "Criteria Entanglement Bias" (where judges assign correlated scores across dimensions) and an "Attention limit phenomenon" (where error detection degrades as criteria increase). This confirms that forcing judges to evaluate multiple dimensions simultaneously leads to diluted attention and missed critical issues.
>
> **2. Prompt Design vs. Multi-Round Structure**
>
> **Our ablation study (Table 2) reveals that neither is sufficient on its own; the performance gain stems specifically from the sequential dependency between the two.** We found that "Branching Only" (Turn 1 alone) leads to significant drops because identifying issues without verifying them causes over-sensitivity to minor flaws. Conversely, "Unconditioned Rethinking" (Turn 2 without Turn 1's focus) also underperforms the full model because without a directed focus, the second pass suffers from the same diffusion as the baseline. Thus, the architectural split – using Turn 1 to prune the search space and Turn 2 to verify the hypotheses – is the key driver of performance.
>
>
> > **Q2.1 Scaling the Number of Turns**
>
> We investigated whether extending the "think twice" principle to "think more times" (or more) yields further gains. We trained variants of our model with 3 and 4 turns, where additional turns use the same prompt as Turn 2. Note that we recognize that this may not be the optimal prompting strategy for 3 or 4 turns, and we leave a more systematic exploration of prompting strategies to future work. For the reward design, we adopt the same binary outcome reward to the last turn and the format reward for each turn to avoid the incomplete intermediate outputs.
>
> **Experiment Results (Accuracy):**
>
> **(1) Model: Qwen3-8B**
>
> | Turn Type | RewardBench | RMBench | RMB |
> | :--- | :--- | :--- | :--- |
> | 1 Turn | 90.1 | 82.6 | 67.4 |
> | 2 Turns (Ours) | **91.0** | **85.0** | **71.8** |
> | 3 Turns (New) | 90.8 | 84.3 | 70.1 |
> | 4 Turns (New) | 88.5 | 82.4 | 68.3 |
>
> **(2) Model: Qwen3-14B**
>
> | Turn Type | RewardBench | RMBench | RMB |
> | :--- | :--- | :--- | :--- |
> | 1 Turn | 90.9 | 83.2 | 71.0 |
> | 2 Turns (Ours) | **92.1** | **85.9** | **74.7** |
> | 3 Turns (New) | 91.5 | 85.2 | 73.1 |
> | 4 Turns (New) | 89.8 | 84.6 | 71.9 |
>
> **Insight**: The results indicate that, in this setting, **two turns represent an optimal trade-off**. Performance peaks at 2 turns and degrades as turns increase. We hypothesize that beyond two turns, the model begins to "over-think," hallucinating subtle errors during repeated verification or drifting away from the original context. This validates our design choice: one turn for hypothesis generation (Branching) and one for verification (Rethinking) is sufficient and robust.

---

> ### Author Response · Authors · 2025-11-22
> **Response to Reviewer 4rrV (2/2)**
>
> > **Q2.2 Justifying Subtask Design**
>
> To validate our design, we implemented the suggested baseline: **"Weighted-rubric scoring"**. In this setup, the model generates importance weights for rubrics given the prompt, generates independent scores for each rubric, and aggregates them.
>
> **Implementation Details:**
>
> To ensure a fair comparison, we trained this baseline using the same backbone (Qwen3-8B) and the same reinforcement learning framework (GRPO) as our main model. The implementation followed a structured three-step process within the generation window:
>
> Weight Generation: The model is prompted to output a weight for each of the 9 evaluation dimensions defined in our paper, where all weights should summed to 1. If the number of output weights is not 9 or not summed up to 1, it would lead to a format penalty as designed in the paper.
> Independent Scoring: The model then generates a concise rationale and a binary score for every dimension.
>
> Aggregation: The final reward score is computed as the weighted sum. During training, the GRPO reward signal was applied based on whether this final weighted score correctly ranked the ground-truth preferred response. We maintained the same maximum token budget (16,384 tokens) as the BR-RM to control for test-time compute.
>
> **Experiment Results:**
>
> **(1) Model: Qwen3-8B**
>
> | Method | RewardBench | RMBench | RMB |
> | :--- | :--- | :--- | :--- |
> | Ours | **91.0** | **85.0** | **71.8** |
> | Weighted-rubric scoring (new) | 90.5 | 83.6 | 68.6 |
>
> **(2) Model: Qwen3-14B**
>
> | Method | RewardBench | RMBench | RMB |
> | :--- | :--- | :--- | :--- |
> | Ours | **92.1** | **85.9** | **74.7** |
> | Weighted-rubric scoring (new) | 91.5 | 84.1 | 72.5 |
>
>
> **Insight**: **Our sequential method outperforms the weighted/parallel approach.** The weighted approach essentially re-introduces judgment diffusion. By requiring the model to generate rationales for multiple rubrics in parallel (even with weights), we force it to spread its compute budget. In contrast, our Branch-and-Rethink framework effectively "prunes" the evaluation tree in Turn 1 , allowing Turn 2 to dedicate all reasoning capacity to scrutinizing "only what matters most". This confirms that the sequential dependency – where the analysis is conditioned on the selected focus – is critical to performance. That being said, we agree with the Reviewer that “the possibilities of designs are endless”, and there may exist other, even more performant designs that are yet to be discovered.

---

### Official Review · Reviewer_7A17 · 2025-10-31

**Soundness:** 3
**Presentation:** 4
**Contribution:** 3
**Rating:** 6
**Confidence:** 3

**Summary:**

This paper diagnoses "judgment diffusion" in existing Reward Models (RMs), where attention is spread thinly across many quality dimensions, leading to shallow analysis. It introduces Branch-and-Rethink (BR-RM), a two-turn framework that first performs Adaptive Branching to select a few instance-critical dimensions and sketch hypotheses, followed by Branch-Conditioned Rethinking for a targeted, second-look analysis based on the initial findings . The model is trained with GRPO-style RL using a simple binary outcome reward and achieves state-of-the-art (SOTA) performance on three challenging RM benchmarks: RewardBench, RM-Bench, and RMB.

**Strengths:**

1. The paper's core idea is reasonable. Diagnosing "judgment diffusion" and transferring the "think-twice" principle from solvers (LLMs) to judges (RMs) is a clever and logical contribution.

2. The paper is supported by strong SOTA results across three diverse benchmarks and is validated by exceptionally comprehensive ablation studies that justify each design component.

3. The paper is well-written and clearly structured. The core problem and the proposed solution are easy to understand.

4. This work addresses a critical bottleneck in LLM alignment by creating RMs that are more sensitive to subtle yet consequential errors, which is essential for developing more reliable AI systems.

**Weaknesses:**

1. Cost: The BR-RM is a two-stage generative model. Compared to a scalar RM, which requires a single forward pass, this approach introduces substantial latency and complexity, especially during RLHF training where the RM is called millions of times. The paper doesn't quantify this two-turn cost, making its practical viability for large-scale application questionable.

2.  The method relies heavily on a predefined "universal set of criteria" and "task-specific evaluation hierarchies". The performance seems contingent on these human-designed sets, making the approach potentially brittle. If a critical evaluation dimension for a new task is missing from the universal set of criteria, will the model be unable to "branch" to it and fail?

3. The paper attempts to differentiate BR-RM from existing Reasoning RMs, but the distinction feels minor. The core contribution appears to be a strong engineering improvement and a clever prompting strategy rather than a fundamental conceptual leap over prior works.

**Questions:**

1. See weakness 1

2. See weakness 2

---

> ### Author Response · Authors · 2025-11-22
> **Response to Reviewer 7A17 (1/2)**
>
> We thank the reviewer for their constructive feedback and positive assessment. We appreciate your recognition that our diagnosis of "judgment diffusion" is reasonable and that our "think-twice" framework is a logical contribution supported by strong SOTA results and comprehensive ablation studies. To address your concerns regarding cost, we provided a detailed latency analysis. We also clarified the robustness of our universal criteria set as a semantic basis and further distinguished our architectural novelty from standard reasoning models.
>
>
> > **W1: Cost and Latency**
>
> We acknowledge the reviewer's concern regarding the inference cost of a two-stage generative model. To address this specifically, we conducted a controlled latency benchmark on 2*H100 GPU, comparing our model against both a standard Scalar RM and a concurrent Reasoning RM (RM-R1) of similar size.
> **Latency vs. Accuracy Trade-off:**
> | Model | Type | Latency (s) / instance | Avg Accuracy |
> | :--- | :--- | :--- | :--- |
> | Internlm2-7b-reward | Scalar | 0.01s | 72.3% |
> | INF-ORM-Llama3.1-70B | Scalar | 0.25s | 78.8% |
> | RM-R1-Qwen-Instruct-7B | Generative (One-Turn) | 5.1s | 73.9% |
> | RM-R1-Qwen-Instruct-32B | Generative (One-Turn) | 10.3s | 81.2% |
> | BR-RM-Qwen-8B (Ours) | Generative (Two-Turn) | 9.5s | 82.6% |
> | BR-RM-Qwen-14B (Ours) | Generative (Two-Turn) | 15.9s | 84.2% |
>
> **Analysis:**
> **1. Efficiency compared to Reasoning Baselines**: Our results demonstrate that **"Thinking Twice" is more efficient than scaling parameters.**
> Our **8B Two-Turn model** (9.5s) is actually **faster** than the **32B One-Turn baseline** (10.3s) while achieving higher accuracy (82.6% vs 81.2%).
> This confirms that allocating test-time compute to a focused second pass (our method) yields better returns than simply increasing model size or generating a single broad critique.
>
> **2. Quality vs. Speed Trade-off against Scalar RMs**: While scalar models are significantly faster (0.01s - 0.25s), they hit a performance ceiling. The strongest scalar baseline (70B) trails our 8B model by **3.8 points** (78.8% vs 82.6%). In RLHF, the quality of the reward signal is often the bottleneck; a noisy or exploitable reward signal (common in lower-accuracy models) can destabilize alignment regardless of generation speed. We argue that the increased latency is a justifiable investment for high-stakes preference labeling (e.g., Rejection Sampling or Offline RL), where accuracy is paramount.
>
> In addition, although a slower RM does slow down RLHF, **this slowdown isn’t as dramatic in practice** as the slower generation of a Reasoning RM (that could be ~1000x slower compared to a Scalar RM) might suggest. This is **because RLHF also requires generating responses from the policy being trained, which is already quite time-consuming**, especially when training reasoning models. This is even less of an issue with asynchronous RL, a paradigm currently gaining traction in the community that can decouple policy rollouts and reward generation from model updates, thus avoiding “idle time” while waiting for all generations to complete.
>
>
>
>
> > **W2: Brittleness of Predefined Criteria**
>
> We appreciate this thoughtful critique. However, we argue that the "universal set" is not a liability but a necessary architectural constraint that solves the "Judgment Diffusion" problem.
>
> **1. Dimensions as a Semantic "Basis Set"**: We view the nine dimensions (e.g., Logical Reasoning, Instruction Adherence, Safety Awareness) not as a rigid checklist, but as a semantic basis set that spans the latent space of response quality. LLMs excel at semantic projection; they do not require an exact keyword match to identify an error. Instead, they dynamically map novel failure modes into the nearest high-level cognitive bucket. For instance, a "tone violation" (missing from the list) semantically projects onto "Communication Clarity"; a "negative constraint violation" projects onto "Instruction Adherence". This flexibility allows the fixed set to cover most failure modes without needing an infinite list.
>
> **2. Constraint prevents Focus Dilution (The "Why")**: The core insight of our work is that unconstrained evaluation leads to "Focus Dilution" . As shown in our preliminary study (Figure 1), when models are left to define their own criteria, they often hallucinate relevance in low-priority areas (e.g., discussing "Writing Clarity" for a coding task) . By constraining the model to select from a high-level universal set, we force a "competition" for attention. This regularizes the reasoning process, ensuring the model commits its compute budget to the most salient type of error (e.g., "Implementation Capability" for code) rather than drifting into shallow, unstructured critique.

---

> ### Author Response · Authors · 2025-11-22
> **Response to Reviewer 7A17 (2/2)**
>
> > **W3: Novelty of the proposed approach**
>
> We respectfully disagree that the distinction is minor. While prior Reasoning RMs introduce "thinking," they typically treat reasoning as an unconstrained "stream of consciousness," which we show leads to **Judgment Diffusion** – spreading compute across irrelevant criteria (e.g., discussing "Writing Clarity" on code tasks). Our conceptual leap is restructuring judgment from generative exploration to hypothesis-driven verification. **By architecturally enforcing a "Pruning" step (Turn 1) followed by a "Conditioned" step (Turn 2), we solve the resource allocation problem that standard CoT fails to address**. This is confirmed by our ablation study: a "Single-Turn Model" (which uses the exact same prompting content but merges the steps) significantly underperforms our two-turn architecture, proving that **the performance gains stem from the structural dependency of the two passes, not just "clever prompting."**

---

### Author Response · Authors · 2025-11-22
**General Response**

Dear Reviewers,

We are deeply grateful for your insightful feedback and valuable suggestions. Your comprehensive reviews have provided us with essential guidance to enhance our work. We would like to start by expressing our appreciation for the positive recognition of the strengths of our study, including:

- **Problem Insight**: The diagnosis of "judgment diffusion" and "focus dilution" is recognized as a sound and critical observation that addresses a key bottleneck in LLM alignment. (`f3LS`, `4rrV`, `7A17`)

- **Methodological Novelty**: The proposed "Branch-and-Rethink" framework is considered a clever, logical, and intuitive transfer of the "think-twice" principle from solvers to judges. (`4rrV`, `7A17`)

- **SOTA Performance**: The model achieves the best average accuracy among compared methods, demonstrating strong state-of-the-art performance across diverse benchmarks (RewardBench, RM-Bench, and RMB). (`f3LS`, `4rrV`, `7A17`)

- **Experimental Rigor**: The ablation studies are praised as exceptionally comprehensive, effectively isolating specific contributions and supporting our core design claims. (`4rrV`, `7A17`)

- **Objective Design**: The combination of strict format penalties and binary outcome rewards is highlighted as an effective, easy-to-implement design that aligns perfectly with the evaluation objective. (`4rrV`)

We have responded individually to each reviewer’s questions and conducted additional experiments to address your concerns. Below is a summary of the key updates and clarifications included in our responses:

- **New Experiments on Scaling**: Conducted supplementary experiments on scaling the number of turns (3 and 4 turns) to validate the optimality of the two-turn design (`f3LS`).

- **New Baseline Comparison**: Implemented and evaluated the "Weighted-Rubric Scoring" baseline to justify our specific subtask design (`f3LS`).

- **Efficiency Analysis**: Provided a detailed latency and efficiency comparison between our two-turn model, scalar RMs, and concurrent reasoning RMs (`7A17`).

- **Focus Dilution Insight**: Elaborated on the theoretical underpinnings of "Focus Dilution" and the cognitive bottleneck in single-pass models (`f3LS`).

- **Updated Comparisons**: Completed the benchmark tables with missing cells for external baselines to ensure a fully reproducible comparison (`4rrV`).

- **Reward Dynamics**: Clarified that the format reward acts as a constraint while the outcome reward incentivizes reasoning depth, preventing mode collapse to short traces (`4rrV`).

- **Dimension Definitions**: Expanded on the definitions, coverage, and adaptive nature of the nine evaluation dimensions to address concerns about brittleness (`4rrV`, `7A17`).

- **Cost Analysis**: Clarified the training costs and ensured fair comparisons regarding compute budgets (`4rrV`, `7A17`).

We sincerely thank you again for your contributions to improving our work. If there are any additional concerns or questions, we are fully prepared to address them.

---

### Meta-Review · Area_Chair_XiLU · 2026-01-06

**Summary:**

This paper proposes branch-and-rethink reward model designed to mitigate judgment diffusion through a two-stage process. Reviewers broadly agree that judgment diffusion is a real issue and that the empirical results are competitive. However, there are consistent concerns across reviews, including limited novelty and insight, computational cost of the method, and the limited generality. Reviewer 7A17 and f3LS had concerns that the conceptual difference from existing reasoning reward models is modest and that the paper lacks in-depth analysis. Reviewers 4rrV and 7A17 raised concerns about the cost of using slower RM, and the argument that generating responses from the policy being trained is already time-consuming is not fully convincing. Reviewers also consistently highlighted that the reliance on heuristic design reduces generality. Although the rebuttal has addressed some points, these concerns remained influential in my overall assessment.

**Reviewer Concerns:**

Reviewer 7A17:
- Cost, latency, and complexity.
- Limited generality due to much human designed criteria and task-specific evaluation hierarchies: may make the method brittle across new tasks.
- Limited novelty.

Reviewer f3LS:
- Lack of insights.
- Too much inductive bias (also raised by Reviewer 7A17), which limits generality.

Reviewer 4rrV:
- Missing baseline results and uneven comparisons. Authors have provided additional results, which partially address this concern.
- Cost and compute transparency (similar concern raised by Reviewer 7A17).
- Limited generalization of reward design (similar concern raised by Reviewer 7A17).

**Reviewer Scores:**

The AC’s best guess is that all reviewers would have kept their original evaluations.

---

### Decision · Program_Chairs · 2026-01-26

Reject